# Offline Reinforcement Learning with OOD State Correction and OOD Action Suppression

**Yixiu Mao**[1], **Qi Wang**[1], **Chen Chen**[1], **Yun Qu**[1], **Xiangyang Ji**[1]
[1]Department of Automation, Tsinghua University
myx21@mails.tsinghua.edu.cn, xyji@tsinghua.edu.cn

## Abstract

In offline reinforcement learning (RL), addressing the out-of-distribution (OOD) action issue has been a focus, but we argue that there exists an OOD state issue that also impairs performance yet has been underexplored. Such an issue describes the scenario when the agent encounters states out of the offline dataset during the test phase, leading to uncontrolled behavior and performance degradation. To this end, we propose SCAS, a simple yet effective approach that unifies OOD state correction and OOD action suppression in offline RL. Technically, SCAS achieves value-aware OOD state correction, capable of correcting the agent from OOD states to high-value in-distribution states. Theoretical and empirical results show that SCAS also exhibits the effect of suppressing OOD actions. On standard offline RL benchmarks, SCAS achieves excellent performance without additional hyperparameter tuning. Moreover, benefiting from its OOD state correction feature, SCAS demonstrates enhanced robustness against environmental perturbations.

## 1 Introduction

Deep reinforcement learning (RL) shows promise in solving sequential decision-making problems, gaining increasing interest for real-world applications [42, 57, 63, 53, 7]. However, deploying RL algorithms in extensive scenarios poses persistent challenges, such as risk-sensitive exploration [13] and time-consuming episode collection [27]. Recent advances view offline RL as a hopeful solution to these challenges [34]. Offline RL aims to learn a policy from a fixed dataset without further interactions [32]. It can tap into existing large-scale datasets for safe and efficient learning [23, 37, 50].

In offline RL research, a well-known concern is the out-of-distribution (OOD) action issue: the evaluation of OOD actions causes extrapolation error [12], which can be exacerbated by bootstrapping and result in severe value overestimation [34]. To address this issue, a large body of work has emerged to directly or indirectly *suppress OOD actions* during training, employing various techniques such as policy constraint [12, 30, 10], value penalization [31, 2, 6], and in-sample learning [29, 14, 71].

Distinguished from most previous works, this paper argues that, apart from the OOD action issue, there exists an *OOD state issue* that also impairs performance yet has received limited attention in the field. Such an issue refers to the agent encountering states out of the offline dataset during the policy deployment phase (i.e., test phase). The occurrence of OOD states can be attributed to OOD actions, stochastic environments, and real-world perturbations. Since typical offline RL algorithms do not involve policy training in OOD states, the agent tends to behave in an uncontrolled manner once entering OOD states in the test phase. This can further exacerbate the state deviation from the offline dataset and lead to severe degradation in performance [34, 75].

In mitigating this OOD state issue, existing limited work attempts to train the policy to correct the agent from OOD states to in-distribution (ID) states [75, 22]. Technically, Zhang et al. [75] construct a dynamics model and a state transition model and align them to guide the agent to ID regions, while

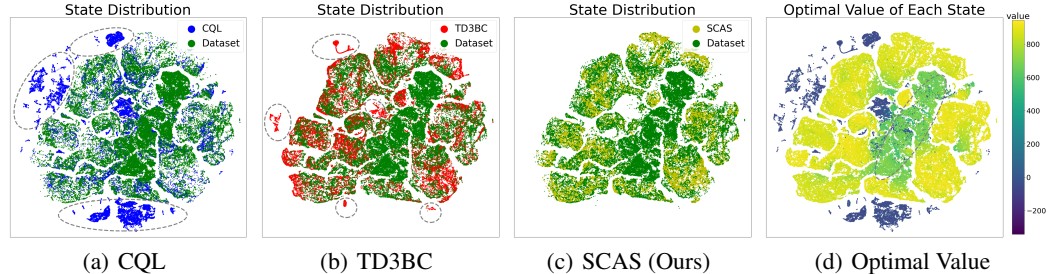

| (a) CQL | (b) TD3BC | (c) SCAS (Ours) | (d) Optimal Value |

Figure 1: **The resulting state distributions of offline RL algorithms and optimal values of states.** (a,b,c) The state distributions generated by the learned policies of various algorithms compared with that of the offline dataset on halfcheetah-medium-expert. (d) The corresponding optimal value of each state, which is obtained by running TD3 online to convergence. *SCAS-induced state distribution is almost entirely within the support of the offline distribution and avoids the low-value areas*, while CQL and TD3BC tend to produce OOD states with extremely low values.

Jiang et al. [22] resort to an inverse dynamics model for policy constraint. However, they deal with the OOD state and OOD action issues separately, requiring extra OOD action suppression components and complex distribution modeling, which sacrifices computational efficiency and algorithmic simplicity. Moreover, correcting the agent to all ID states impartially could be problematic, especially when the dataset contains substantial suboptimal states. As a result, the performance of prior methods also leaves considerable room for improvement.

In this paper, we aim to address these two fundamental OOD issues simultaneously by proposing a simple yet effective approach for offline RL. We term our method SCAS due to its integration of OOD **S**tate **C**orrection and OOD **A**ction **S**uppression. We start with solving an analytical form of a value-aware state transition distribution, which is within the dataset support but skewed toward high-value states. Then, we align it with the dynamics induced by the trained policy on perturbed states via KL divergence. This operation intends to correct the agent from OOD states to high-value ID states, a concept we refer to as *value-aware* OOD state correction. Through some derivations, it also eliminates the necessity of training a multi-modal state transition model. Furthermore, we show theoretically and empirically that, while designed for OOD state correction, SCAS regularization also exhibits the effect of OOD action suppression. We evaluate SCAS on the offline RL benchmarks including D4RL [9] and NeoRL [49]. SCAS achieves excellent performance with consistent hyperparameters without additional tuning. Moreover, benefiting from its OOD state correction ability, SCAS demonstrates improved robustness against environmental perturbations.

To summarize, the main contributions of this work are:

- We systematically analyze the underexplored OOD state issue in offline RL and propose a simple yet effective approach SCAS *unifying OOD state correction and OOD action suppression*.
- Our approach achieves *value-aware* OOD state correction, which circumvents modeling complex distributions and significantly improves performance over vanilla OOD state correction methods.
- Empirically[1], our approach demonstrates superior performance on standard offline RL benchmarks and enhanced robustness in perturbed environments *without additional hyperparameter tuning*.

## 2 Preliminaries

In reinforcement learning, we generally characterize the environment as a Markov Decision Process (MDP) $\mathcal{M} = (\mathcal{S}, \mathcal{A}, P, R, \gamma, d_0)$, with state space $\mathcal{S}$, action space $\mathcal{A}$, transition dynamics $P : \mathcal{S} \times \mathcal{A} \rightarrow \Delta(\mathcal{S})$, reward function $R : \mathcal{S} \times \mathcal{A} \rightarrow \mathbb{R}$, discount factor $\gamma \in [0, 1)$, and initial state distribution $d_0$ [61]. The agent interacts with the environment and seeks a policy $\pi : \mathcal{S} \rightarrow \Delta(\mathcal{A})$ to maximize the expected discounted return $\eta(\pi)$:

$$\eta(\pi) = \mathbb{E}_{s_0 \sim d_0, a_t \sim \pi(\cdot|s_t), s_{t+1} \sim P(\cdot|s_t, a_t)} \left[ \sum_{t=0}^{\infty} \gamma^t R(s_t, a_t) \right]. \tag{1}$$

---

[1]Our code is available at https://github.com/maoyixiu/SCAS.

For any policy $\pi$, we define the value function as $V^\pi(s) = \mathbb{E}_\pi \left[ \sum_{t=0}^\infty \gamma^t R(s_t, a_t) | s_0 = s \right]$ and the state-action value function ($Q$-value function) as $Q^\pi(s, a) = \mathbb{E}_\pi \left[ \sum_{t=0}^\infty \gamma^t R(s_t, a_t) | s_0 = s, a_0 = a \right]$.

**Offline RL.** In offline RL, the agent can only access a static dataset $\mathcal{D} = \{(s_t^i, a_t^i, s_{t+1}^i, r_t^i)\}$. We denote the empirical behavior policy of $\mathcal{D}$ by $\beta(a|s)$ and the empirical dynamics model by $M(s'|s, a)$, both of which depict the conditional distributions observed in the dataset [12]. Typical actor-critic algorithms [56, 18] evaluate policy $\pi$ by minimizing Bellman loss:

$$L_Q(\theta) = \mathbb{E}_{(s,a,s')\sim\mathcal{D}}[(Q_\theta(s, a) - R(s, a) - \gamma\mathbb{E}_{a'\sim\pi_\phi(\cdot|s')}Q_{\theta'}(s', a'))^2], \tag{2}$$

where $\pi_\phi$ and $Q_\theta$ are the parameterized policy and $Q$ function, and $Q_{\theta'}$ is a target network whose parameters are updated via Polyak averaging [42].

Simultaneously, policy improvement in policy iteration is achieved via maximizing the Q-value:

$$L_\pi(\phi) = -\mathbb{E}_{s\sim\mathcal{D}, a\sim\pi_\phi}[Q_\theta(s, a)]. \tag{3}$$

**OOD action issue.** In offline RL, *OOD actions* refer to actions outside the support of the behavior policy $\beta(\cdot|s)$ at a specific state $s \in \mathcal{D}$. Since the Q-values of OOD actions can be poorly estimated and the policy improvement is towards maximizing the estimated $Q_\theta$, the resulting policy tends to prioritize the OOD actions with overestimated values, leading to poor performance [12].

## 3 OOD State Correction

The following focuses on the OOD state issue and OOD state correction in offline RL. In Section 3.1, we systematically analyze the OOD state issue, introduce the concept of OOD state correction, and point out limitations of prior methods. Then we present the proposed approach SCAS in Section 3.2.

### 3.1 OOD State Issue in Offline RL

In offline RL, *OOD states* refer to states not in the offline dataset. The OOD state issue (Definition 1) pertains to scenarios where the agent enters OOD states during the test phase, potentially resulting in catastrophic failure [34]. However, such a topic is rarely investigated in the literature, and existing studies lack deep insights. We mathematically formulate the OOD state issue as follows.

**Definition 1** (OOD state issue). *There exists $s \in \mathcal{S}$, such that $d^\pi_{\mathcal{M}_\mathcal{T}}(s) > 0$ and $d_\mathcal{D}(s) = 0$, where $\mathcal{M}_\mathcal{T}$ is the MDP of the test environment, $\pi$ is any learned policy, $d^\pi_{\mathcal{M}_\mathcal{T}}$ is the state probability density induced by $\pi$ in $\mathcal{M}_\mathcal{T}$, and $d_\mathcal{D}$ is the state probability density in the offline dataset.*

**Origins and consequence of OOD states.** During the test phase, the OOD states occur primarily in three scenarios: (i) OOD actions: the learned policy, not perfectly constrained within the support of the behavior policy, executes unreliable OOD actions, leading to OOD states. (ii) Stochastic environment: the initial state of the actual environment may fall outside the offline dataset. In addition, stochastic dynamics can also lead to states outside the dataset, even when taking ID actions in ID states. (iii) Perturbations: commonly seen in real-world robot applications, some unexpected perturbations can propel the agent into OOD states (e.g., wind, human interference).

During offline training, the typical Bellman updates involve only ID states, and the policies in OOD states are not trained. As a result, when encountering OOD states in the test phase, the agent would exhibit uncontrolled behavior, and the state deviation from the offline dataset can be further exacerbated over time steps, severely degrading performance [34].

**OOD state correction.** To mitigate this OOD state issue, an intuitive solution is to train a policy capable of correcting the agent from OOD states to ID states, a concept known as *OOD state correction* [75]. Specifically, during offline training, we can perturb the original state $s$ in the dataset into $\hat{s}$ to generate substantial OOD states. Then consider the scenario where the agent starts from $\hat{s}$, follows the trained policy $\pi$, and transitions to the next state $\hat{s}'$. To reduce state deviation, $\hat{s}'$ is expected to be close to the offline dataset. Thus we can align the distribution of $\hat{s}'$ with an ID state distribution to regularize the policy and achieve OOD state correction.

Continuing the above train of thought, SDC [75] generates the ID state distribution by feeding the original state $s$ into a trained state transition model $N(s'|s)$ of the dataset. This model characterizes

the conditional state transition distribution in the dataset and is implemented by a conditional variational auto-encoder (CVAE) [58]. After pretraining a dynamics model $M(s'|s, a)$ and the state transition model $N(s'|s)$, SDC introduces the following policy regularizer for OOD state correction:

$$\min_{\pi} \mathop{\mathbb{E}}_{s \sim \mathcal{D}} \mathop{\mathbb{E}}_{\hat{s} \sim \mathcal{N}_{\sigma}(s)} \left[ \text{MMD}(M(\cdot|\hat{s}, \pi(\cdot|\hat{s})), N(\cdot|s)) \right], \tag{4}$$

where $\hat{s}$ is a Gaussian noise perturbed version of the original state $s$, $\sigma$ is the standard deviation of the Gaussian, $M(\cdot|\hat{s}, \pi(\cdot|\hat{s}))$ is shorthand for $\mathbb{E}_{\hat{a} \sim \pi(\cdot|\hat{s})} M(\cdot|\hat{s}, \hat{a})$, and MMD is the maximum mean discrepancy measure. More recently, OSR [22] directly aligns the trained policy distribution at the perturbed state with a CVAE inverse dynamics model to constrain the policy in OOD states.

**Limitations.** However, the regularizers of prior methods are only designed to deal with this OOD state issue. To mitigate OOD actions, they require an additional conservative Q learning (CQL) term [31] in value estimation to penalize Q-values of OOD actions. In addition, the state transition distribution and the inverse dynamics distribution are multi-modal in many scenarios [43]. The necessity of extra OOD action suppression components and complex distribution modeling compromises their computational efficiency and algorithmic simplicity. Moreover, correcting the agent to all ID states impartially could be problematic, particularly when the offline dataset contains a large portion of suboptimal states. In such cases, vanilla OOD state correction can lead to suboptimal behaviors. Consequently, there is also significant potential for improvement in the performance of prior methods.

For a more comprehensive discussion of related work, please refer to Appendix A.

## 3.2 Value-aware OOD State Correction

The objective of this work is to formulate a simple yet effective policy regularizer for offline RL that unifies OOD state correction and OOD action suppression. Moreover, we aim to achieve *value-aware* OOD state correction, involving the correction of the agent from OOD states to high-value ID states.

**Value-aware state transition.** For the ID state distribution to which the agent is corrected, we expect a value-aware state transition distribution $N^*(\cdot|s)$ that lies within the support of the dataset state transition distribution $N(\cdot|s)$ but is skewed toward high-value states $s'$. To ensure stability and, more importantly, to enable our subsequently designed algorithm to circumvent modeling complex distributions, we seek a soft optimal version of it. To this end, we consider the following problem[2]:

$$\max_{N^*} \mathop{\mathbb{E}}_{s \sim \mathcal{D}} \left[ \alpha \mathop{\mathbb{E}}_{s' \sim N^*(\cdot|s)} V(s') - D_{\text{KL}}(N^*(\cdot|s) \| N(\cdot|s)) \right], \tag{5}$$

where $\alpha$ is a hyperparameter to balance the two terms.

The optimization problem above has a closed-form solution:

$$N^*(s'|s) = \frac{1}{Z(s)} \exp\left(\alpha V(s')\right) N(s'|s), \tag{6}$$

where $Z(s) = \sum_{s'} \exp\left(\alpha V(s')\right) N(s'|s)$ is a normalization factor. It can be seen from Eq. (6) that $\text{supp}(N^*(\cdot|s)) \subseteq \text{supp}(N(\cdot|s))$. Note that $\alpha$ is a key hyperparameter that controls the significance of the values of next states in SCAS's OOD state correction. As $\alpha$ increases, $N^*(\cdot|s)$ becomes more skewed toward the optimal $s'$ in the support of $N(\cdot|s)$.

**OOD state correction.** In order to produce substantial OOD states, we perturb each state $s \in \mathcal{D}$ with Gaussian noise $\mathcal{N}(0, \sigma^2)$, resulting in perturbed state $\hat{s}$. It is worth noting that the dataset used for RL training remains unchanged. We perturb the states solely to formulate the regularizer.

We anticipate the following value-aware OOD state correction scenario, where the agent starts from OOD state $\hat{s}$, follows the trained policy $\pi$, and transitions to the high-value ID state $s'$ in the distribution of $N^*(\cdot|s)$. To this end, we train the policy $\pi$ to align the dynamics induced by $\pi$ on the perturbed state $\hat{s}$ with the value-aware state transition distribution at the original state $s$ via KL divergence. That is, we regularize $\pi$ by minimizing:

$$\min_{\pi} \mathop{\mathbb{E}}_{s \sim \mathcal{D}} \mathop{\mathbb{E}}_{\hat{s} \sim \mathcal{N}_{\sigma}(s)} D_{\text{KL}}(N^*(\cdot|s) \| M(\cdot|\hat{s}, \pi(\cdot|\hat{s}))). \tag{7}$$

---

[2]Note that the regularizer $D_{\text{KL}}(N^*(\cdot|s) \| N(\cdot|s))$ can constrain the support of $N^*(\cdot|s)$ within that of $N(\cdot|s)$, because if $\text{supp}(N^*(\cdot|s)) \not\subseteq \text{supp}(N(\cdot|s))$ at some state $s$, then $D_{\text{KL}}(N^*(\cdot|s) \| N(\cdot|s)) = \infty$.

By substituting the analytical solution of $N^*$ from Eq. (6) into the KL divergence, we have

$$\operatorname*{argmin}_{\pi} \mathrm{D_{KL}}(N^*(\cdot|s)\|M(\cdot|\hat{s},\pi(\cdot|\hat{s}))) = \operatorname*{argmax}_{\pi} \mathbb{E}_{s'\sim N(\cdot|s)}\left[\frac{\exp\left(\alpha V\left(s'\right)\right)}{Z(s)}\log M(s'|\hat{s},\pi(\cdot|\hat{s}))\right].$$

Note that $N$ is the state transition distribution in the dataset, and $s\sim\mathcal{D}, s'\sim N(\cdot|s)$ is equivalent to $(s,s')\sim\mathcal{D}$. Thus minimizing Eq. (7) is equivalent to maximizing following regularizer:

$$R(\pi) = \mathbb{E}_{(s,s')\sim\mathcal{D}}\mathbb{E}_{\hat{s}\sim\mathcal{N}_\sigma(s)}\left[\frac{\exp\left(\alpha V\left(s'\right)\right)}{Z(s)}\log M(s'|\hat{s},\pi(\cdot|\hat{s}))\right]. \tag{8}$$

As a result, $R(\pi)$ effectively eliminates the need for a pre-trained multi-modal state transition model ($N$ or $N^*$) and enables direct sampling from the dataset for optimization.

However, the normalization factor $Z(s)$ in $R(\pi)$ can be challenging to compute. We note that the regularizer $R(\pi)$ is derived from the minimization of the KL divergence in Eq. (7). Since we aim to minimize this KL at every state $s$ in $\mathcal{D}$ and $Z(s)$ only affects the relative weights at different $s$, it matters less to precisely restore the correct state weights in $\mathcal{D}$ by computing $Z(s)$, which is empirically hard to estimate and may bring more instability. Thus, we replace $Z(s)$ in $R(\pi)$ with an empirical normalizer $\exp(\alpha V(s))$ for computational stability:

$$R_1(\pi) = \mathbb{E}_{(s,s')\sim\mathcal{D}}\mathbb{E}_{\hat{s}\sim\mathcal{N}_\sigma(s)}\left[\frac{\exp\left(\alpha V\left(s'\right)\right)}{\exp\left(\alpha V\left(s\right)\right)}\log M(s'|\hat{s},\pi(\cdot|\hat{s}))\right]. \tag{9}$$

We provide further rationale behind this choice of the empirical normalizer in Appendix C.1.

**Tractable optimization.** Now we shift focus to the optimization of $R_1(\pi)$. The expectation with respect to $\pi$ can be moved outside the logarithm by Jensen's inequality:

$$R_1(\pi) \geq \mathbb{E}_{(s,s')\sim\mathcal{D}}\mathbb{E}_{\hat{s}\sim\mathcal{N}_\sigma(s)}\mathbb{E}_{a\sim\pi(\cdot|\hat{s})}\left[\frac{\exp\left(\alpha V\left(s'\right)\right)}{\exp\left(\alpha V\left(s\right)\right)}\log M(s'|\hat{s},a)\right], \tag{10}$$

where the equality holds when $\pi$ is deterministic. In general, it is convenient to maximize the lower bound in Eq. (10) using the reparameterization trick. However, to ensure the equality case in Eq. (10), we opt to train a deterministic policy $\pi$. In this case, we can directly maximize $R_1(\pi)$ by computing the gradient of $\pi$ using automatic differentiation [46].

In contrast to model-based RL methods that typically use the learned dynamics model to roll out multi-step trajectories for policy training [20, 73], our algorithm utilizes the dynamics model to propagate the gradient of policy and regularize policy training, resulting in significantly enhanced computational efficiency. Moreover, the nature of one-step dynamics prediction in our method is advantageous for maintaining relatively high prediction accuracy.

## 4 Analysis of OOD Action Suppression

This section focuses on the OOD action issue and shows that the proposed regularizer also exhibits the effect of *OOD action suppression*. In other words, it can also prevent the policy from taking OOD actions, thereby simultaneously addressing the fundamental OOD action issue in offline RL. In offline RL, OOD actions are exclusively defined on ID states. This is because actor-critic training is limited to ID states, and any actions on OOD states would not affect training and cause the OOD action issue mentioned in Section 2. Consequently, for the analysis of OOD actions, it is essential to consider ID states. We define $\bar{R}, \bar{R}_1$ as the ID state version of $R, R_1$, where $\hat{s} = s$. $\bar{R}$ and $\bar{R}_1$ can be regarded as special cases of $R$ and $R_1$, when $\hat{s}$ sampled from $\mathcal{N}(s,\sigma^2)$ is equal to $s$:

$$\bar{R}(\pi) = \mathbb{E}_{(s,s')\sim\mathcal{D}}\left[\frac{\exp\left(\alpha V\left(s'\right)\right)}{Z(s)}\log M(s'|s,\pi(\cdot|s))\right], \tag{11}$$

$$\bar{R}_1(\pi) = \mathbb{E}_{(s,s')\sim\mathcal{D}}\left[\frac{\exp\left(\alpha V\left(s'\right)\right)}{\exp\left(\alpha V\left(s\right)\right)}\log M(s'|s,\pi(\cdot|s))\right]. \tag{12}$$

The proposed regularizer functions as follows: when the agent encounters OOD states, it drives the agent to choose actions leading to ID states, as discussed in Section 3.2. When the agent is in ID states, the ID state part of it comes into play. In the following, we show that it helps circumvent taking OOD actions by analyzing the maximizer of $\bar{R}, \bar{R}_1$ in tabular MDPs.

**Proposition 1.** *Suppose that the environment dynamics is deterministic, then both $\bar{R}(\pi)$ and $\bar{R}_1(\pi)$ achieve their global maximum at the policy $\pi^*$, where*[3]

$$\pi^*(a|s) = \frac{1}{Z(s)} \exp\left(\alpha V\left(M(s, a)\right)\right)\beta(a|s) \tag{13}$$

*The support of $\pi^*$ is within that of the behavior policy $\beta$:*

$$\operatorname{supp}(\pi^*(\cdot|s)) \subseteq \operatorname{supp}(\beta(\cdot|s)), \ \forall s \sim \mathcal{D} \tag{14}$$

*and $\pi^*$ makes the following equation hold:*

$$N^*(\cdot|s) = M(\cdot|s, \pi^*(\cdot|s)), \ \forall s \sim \mathcal{D} \tag{15}$$

Under the deterministic dynamics condition, Proposition 1 shows that $\pi^*$ is constrained within the support of the behavior policy. Thus, our regularizer helps to keep the policy from taking OOD actions. Moreover, $\pi^*$ is able to exactly align $M(\cdot|s, \pi^*(\cdot|s))$ with $N^*(\cdot|s)$, indicating the guidance of the agent to the high-value ID state distributions.

Furthermore, we show in Proposition 2 that even under stochastic dynamics, the optimization of $\bar{R}$ and $\bar{R}_1$ still yields policies constrained within the support of $\beta$. Hence, SCAS also exhibits the effect of OOD action suppression in this more general scenario.

**Proposition 2.** *When the dynamics is stochastic, the maximizers of both $\bar{R}(\pi)$ and $\bar{R}_1(\pi)$ are constrained within the support of the behavior policy:*

$$\operatorname{supp}(\pi^*(\cdot|s)) \subseteq \operatorname{supp}(\beta(\cdot|s)), \ \forall s \sim \mathcal{D} \tag{16}$$

$$\operatorname{supp}(\pi_1^*(\cdot|s)) \subseteq \operatorname{supp}(\beta(\cdot|s)), \ \forall s \sim \mathcal{D} \tag{17}$$

## 5 Implementation Details

SCAS is easy to implement and we design the practical algorithm to be as simple as possible, retaining algorithmic simplicity and improving computational efficiency.

**Dynamics model.** We employ a deterministic dynamics model $M_\omega$. The loss for training the model is

$$L_M(\omega) = \mathop{\mathbb{E}}_{(s,a,s')\sim\mathcal{D}}[\|M_\omega(s,a) - s'\|_2^2] \tag{18}$$

**Policy improvement.** With a deterministic model, we replace the log-likelihood in $R_1(\pi)$ with mean squared error. It is a common approach in RL algorithms to convert a maximum likelihood estimation problem into a regression problem when dealing with Gaussians with fixed variance [10]. As discussed in Section 3.2, we also adopt a deterministic policy model $\pi_\phi$. Thus, we have the following policy regularizer:

---

**Algorithm 1** SCAS

1: Initialize dynamics model $M_\omega$, policy network $\pi_\phi$, $Q$-network $Q_\theta$, and target $Q$-network $Q_{\theta'}$
2: **// Dynamics Model Training**
3: **for** each gradient step **do**
4:     Update $\omega$ by minimizing $L_M(\omega)$ in Eq. (18)
5: **end for**
6: **// Policy Training**
7: **for** each gradient step **do**
8:     Update $\theta$ by minimizing $L_Q(\theta)$ in Eq. (2)
9:     Update $\phi$ by maximizing $J_\pi(\phi)$ in Eq. (20)
10:     Update target network: $\theta' \leftarrow (1-\tau)\theta' + \tau\theta$
11: **end for**

---

$$R_2(\pi_\phi) = \mathop{\mathbb{E}}_{(s,s')\sim\mathcal{D}} \mathop{\mathbb{E}}_{\hat{s}\sim\mathcal{N}_\sigma(s)} \left[ \frac{\exp\left(\alpha V_\theta\left(s'\right)\right)}{\exp\left(\alpha V_\theta\left(s\right)\right)} \|M_\omega(\hat{s}, \pi_\phi(\hat{s})) - s'\|_2^2 \right], \tag{19}$$

where $V_\theta(s) = Q_\theta(s, \bar{\pi}_\phi(s))$ and $\bar{\pi}_\phi$ means $\pi_\phi$ with detached gradients. Using deterministic policy also simplifies the training process without learning a $V$-function. Combining $R_2(\pi_\phi)$ with the standard policy improvement objective, we update the policy by maximizing:

$$J_\pi(\phi) = (1 - \lambda)\mathbb{E}_{s\sim\mathcal{D}}\left[Q_\theta\left(s, \pi_\phi(s)\right)\right] + \lambda R_2(\pi_\phi), \tag{20}$$

---

[3]Here for clarity, we use the notation $M$ with slightly different meanings in different cases: in the stochastic setting, $M : \mathcal{S} \times \mathcal{A} \to \Delta(\mathcal{S})$; in the deterministic setting, $M : \mathcal{S} \times \mathcal{A} \to \mathcal{S}$.

where $\lambda$ is a hyperparameter to balance the two terms. Additionally, following TD3+BC [10], we also normalize $Q_\theta$ in the first term in each mini-batch to maintain a balanced scale across tasks.

**Overall algorithm.** Putting everything together, we present our final algorithm in Algorithm 1.

# 6   Experiments

In this section, we conduct several experiments to examine the performance and properties of SCAS. Please refer to Appendices D and E for experimental details and additional results.

## 6.1   Empirical Evidence of OOD State Correction and OOD Action Suppression

**OOD state correction.** To examine the OOD state correction ability, we compare the state distributions generated by the learned policies of different algorithms with the state distribution of the offline dataset. In detail, we first train SCAS, CQL [31], and TD3+BC [10], and then collect 50,000 samples by running the trained policies separately. We also sample 50,000 states randomly from the offline dataset for comparison. Figures 1(a) to 1(c) plot the state distributions in halfcheetah-medium-expert [9] with t-SNE [62], and Figure 1(d) visualizes the optimal value of each state. We access these values from the learned value function obtained by running TD3 [11] online to convergence.

In Figures 1(a) and 1(b), we observe that the policies learned by CQL and TD3+BC tend to produce OOD states. As depicted in Figure 1(d), these OOD states have extremely low values, so entering them can be detrimental to performance. In contrast, the state distribution induced by SCAS is almost entirely within the support of the offline distribution, demonstrating the OOD state correction ability of SCAS. Moreover, we also note that in the low-value area of the offline state distribution (the grey circle in Figure 1(d)), SCAS exhibits a very low state density, which could be attributed to SCAS's value-aware OOD state correction. We refer the reader to Appendix E.2 for additional experiments validating the OOD state correction effects.

**OOD action suppression.** We empirically evaluate the OOD action suppression effects through the lens of value estimates. We compare SCAS with three baselines: (1) ordinary off-policy RL which is SCAS with $\lambda = 0$ (all other implementations are the same); (2) SDC [75] without additional CQL [31] term to suppress OOD actions; (3) OSR [22] without additional CQL term. We conduct experiments on D4RL datasets [9]. Since value over-estimation (divergence) is the main consequence and evidence of OOD actions [12], we plot the learned Q-values of SCAS and the baselines in Figure 2.

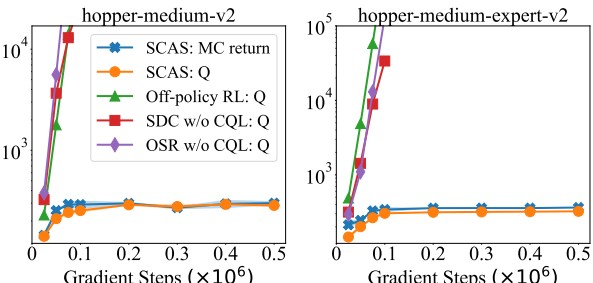

Figure 2: Oracle Q-values of SCAS (estimated by MC return) and learned Q-values of SCAS and other algorithms across optimization steps. Only SCAS's OOD state correction term can achieve OOD action suppression and prevent value over-estimation (divergence).

We also include the oracle Q-values of SCAS by rollouting the trained policy for $1,000$ episodes and evaluating the Monte-Carlo return. Additional results are provided in Appendix E.1.

The results show that the learned Q-values of ordinary off-policy RL, SDC without CQL, and OSR without CQL diverge at early learning stages, suggesting that the algorithms suffer from severe OOD actions. By contrast, the learned Q-values of SCAS stay close to the oracle Q-values. This indicates that SCAS regularization alone is able to suppress OOD actions.

## 6.2   Comparisons on Offline RL Benchmarks

**Tasks.** We evaluate SCAS on D4RL [9] and NeoRL [49] benchmarks. In D4RL, we conduct experiments on Gym locomotion tasks and much more challenging AntMaze tasks. Due to the space limit, *the results on NeoRL* are deferred to Table 4 in Appendix E.3.

Table 1: Averaged normalized scores on Gym locomotion and AntMaze tasks over five random seeds.

| Dataset (v2) | BC | MOPO | OneStep | TD3BC | CQL | IQL | OSR | SDC | SCAS (Ours) |
|---|---|---|---|---|---|---|---|---|---|
| halfcheetah-med | 42.0 | **73.1** | 50.4 | 48.3 | 47.0 | 47.4 | 45.1±0.8 | 45.9±0.5 | 46.6±0.2 |
| hopper-med | 56.2 | 38.3 | 87.5 | 59.3 | 53.0 | 66.2 | 62.0±3.6 | 64.7±3.5 | **102.5±0.3** |
| walker2d-med | 71.0 | 41.2 | **84.8** | **83.7** | 73.3 | 78.3 | **80.1±1.8** | **82.7±1.9** | **82.3±3.0** |
| halfcheetah-med-rep | 36.4 | **69.2** | 42.7 | 44.6 | 45.5 | 44.2 | 43.3±0.2 | 45.1±0.5 | 44.0±0.3 |
| hopper-med-rep | 21.8 | 32.7 | **98.5** | 60.9 | 88.7 | 94.7 | 42.1±12.3 | 94.8±6.5 | **101.6±1.0** |
| walker2d-med-rep | 24.9 | 73.7 | 61.7 | **81.8** | **81.8** | 73.8 | **78.1±1.8** | **78.5±6.0** | **78.1±4.5** |
| halfcheetah-med-exp | 59.6 | 70.3 | 75.1 | **90.7** | 75.6 | 86.7 | 63.7±14.5 | 76.3±5.2 | **91.7±2.7** |
| hopper-med-exp | 51.7 | 60.6 | **108.6** | 98.0 | 105.6 | 91.5 | 78.9±16.4 | 99.9±8.5 | **109.7±3.5** |
| walker2d-med-exp | 101.2 | 77.4 | **111.3** | 110.1 | 107.9 | 109.6 | **108.1±4.4** | **109.2±1.4** | 108.4±3.7 |
| halfcheetah-rand | 2.6 | **35.9** | 2.3 | 11.0 | 17.5 | 13.1 | 1.6±0.1 | 14.2±0.7 | 12.2±3.2 |
| hopper-rand | 4.1 | 16.7 | 5.6 | 8.5 | 7.9 | 7.9 | 3.7±2.6 | 3.1±2.8 | **31.4±0.1** |
| walker2d-rand | 1.2 | 4.2 | **6.9** | 1.6 | 5.1 | 5.4 | -0.1±0.0 | 0.2±0.4 | 1.4±1.1 |
| locomotion total | 472.7 | 593.3 | 735.4 | 698.5 | 708.9 | 718.8 | 606.7 | 714.6 | **810.1** |
| antmaze-umaze | 66.8 | 0.0 | 54.0 | 73.0 | 82.6 | **89.6** | 87.4±5.0 | 81.4±3.8 | **90.4±4.3** |
| antmaze-umaze-div | 56.8 | 0.0 | 57.8 | 47.0 | 10.2 | **65.6** | 55.6±8.0 | 49.6±10.4 | **63.8±16.7** |
| antmaze-med-play | 0.0 | 0.0 | 0.0 | 0.0 | 59.0 | **76.4** | 22.6±7.6 | 55.0±9.6 | **76.6±3.9** |
| antmaze-med-div | 0.0 | 0.0 | 0.6 | 0.2 | 46.6 | 72.8 | 19.6±5.8 | 56.6±10.3 | **80.4±5.4** |
| antmaze-large-play | 0.0 | 0.0 | 0.0 | 0.0 | 16.4 | 42.0 | 0.0±0.0 | 20.8±8.0 | **49.0±4.0** |
| antmaze-large-div | 0.0 | 0.0 | 0.2 | 0.0 | 3.2 | 46.0 | 0.0±0.0 | 25.8±7.5 | **50.6±7.2** |
| antmaze total | 123.6 | 0.0 | 112.6 | 120.2 | 218 | 392.4 | 185.2 | 289.2 | **410.8** |
| runtime | 30m | 900m | 120m | 60m | 250m | 100m | 300m | 420m | 140m |
| hyperparameter tuning | **w/o** | w/ | **w/o** | **w/o** | w/ | **w/o** | w/ | w/ | **w/o** |

**Baselines.** We compare SCAS with prior state-of-the-art offline RL methods as well as the ones specifically designed for OOD state correction, including BC [48], MOPO [73], OneStep RL [5], CQL [31], TD3+BC [10], IQL [29], SDC [75] and OSR [22].

**Hyperparamter tuning.** Offline RL methods are appealing for their ability to generate effective policies without online interaction. Nevertheless, many existing offline RL works involve dataset-specific hyperparameter tuning. The reduction of hyperparameter tuning is crucial for improving practical applicability. In this work, SCAS uses *a single set of hyperparameters for all datasets* in D4RL and NeoRL benchmarks to obtain the reported results.

**Comparisons with baselines.** On D4RL, comparisons of performance, runtime, and hyperparameter tuning information are shown in Table 1. We refer the reader to Appendix E.8 for learning curve details of SCAS. On the Gym locomotion tasks, SCAS outperforms prior methods on most datasets and achieves the highest total score with a single set of hyperparameters. On the challenging AntMaze tasks, SCAS performs better than IQL and outperforms other methods by a very large margin. In NeoRL (Table 4), SCAS performs comparably to MOBILE [59] and outperforms other baselines.

**Runtime.** We present the runtime of algorithms at the bottom of Table 1. SCAS exhibits significantly lower runtime than MOPO, SDC, and OSR and is comparable to other model-free baselines.

**Generality.** SCAS is a generic model-based regularizer that can be easily integrated into existing offline RL algorithms. The corresponding results and analysis are provided in Appendix E.5.

## 6.3 Comparisons in Perturbed Environments

In this section, we evaluate the algorithms in a more real-world setting where the agent receives uncertain perturbations during test time. OOD state correction is even more critical in such scenarios since the agent can enter OOD states after perturbation. To simulate this scenario, we add varying steps of Gaussian noise with a magnitude of $0.5$ to the actions conducted by the policy during test time. Specifically, the policy is trained on standard D4RL datasets but is tested in the perturbed environments. We control the strength of perturbations by adjusting the number of perturbation steps.

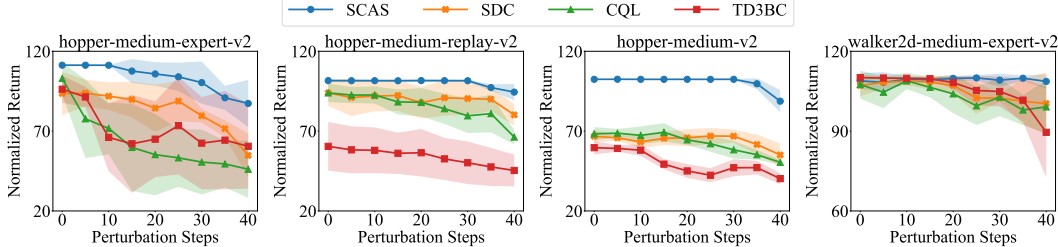

Figure 3: Comparisons in the perturbed environments with varying perturbation levels. The perturbation steps are the steps of Gaussian noise added to the conducted actions in an episode. SCAS exhibits better robustness against environmental perturbations during the test phase.

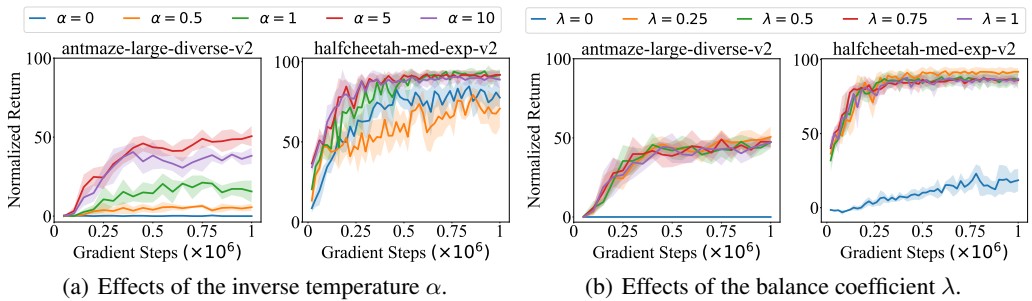

(a) Effects of the inverse temperature $\alpha$.

(b) Effects of the balance coefficient $\lambda$.

Figure 4: Parameter study on the inverse temperature $\alpha$ and the balance coefficient $\lambda$. (a) An appropriately large $\alpha$ is crucial for achieving good performance. (b) The proposed SCAS regularization is essential and demonstrates robustness to changes in $\lambda$.

Figure 3 shows the results of TD3+BC, CQL, SDC, and SCAS on various datasets over five random seeds. We observe that SCAS consistently outperforms previous methods across different perturbation levels and also exhibits less performance degradation against perturbations. Therefore, SCAS enjoys better robustness against perturbations in the complex and unpredictable environments.

## 6.4 Parameter Study

We examine the effects of the inverse temperature $\alpha$, the balance coefficient $\lambda$, and the noise scale $\sigma$. Due to the space limit, *the results for $\sigma$ and on additional datasets* are deferred to Appendix E.6. A sensitivity analysis on dynamics model errors is also provided in Appendix E.7.

**Inverse temperature $\alpha$.** $\alpha$ is the key hyperparameter in SCAS for achieving value-aware OOD state correction. If $\alpha = 0$, the effect degenerates to vanilla OOD state correction. Figure 4(a) displays the learning curves of SCAS with different $\alpha$. The results show that **a large $\alpha$ is *crucial*** for achieving good performance (also verified on more tasks in Figure 6), clearly demonstrating the effectiveness of our value-aware OOD state correction. However, too large $\alpha$ ($\alpha = 10$) induces less satisfying performance, probably due to the increased variance of the learning objective.

**Balance coefficient $\lambda$.** $\lambda$ in Eq. (20) controls the balance between vanilla policy improvement and SCAS regularization. We vary $\lambda$ within the range $[0, 1]$ and present the learning curves of SCAS in Figure 4(b). Notably, SCAS is able to converge to good performance over a very wide range of $\lambda$ (also verified on more tasks in Figure 7). An interesting finding is that even when $\lambda = 1$ and the signal from RL improvement (max Q) is removed, SCAS still performs well on most tasks. This could be attributed to the fact that value-aware OOD state correction implies some sort of improvement in policy by maximizing the values of policy-induced next states.

## 7 Conclusion and Limitations

In this paper, we systematically analyze the OOD state issue in offline RL and propose SCAS, a simple yet effective approach that unifies *OOD state correction* and *OOD action suppression*. SCAS also achieves *value-aware* OOD state correction, significantly improving performance over vanilla

OOD state correction. Empirical results validate the properties of SCAS, showcasing its superior performance on the offline RL benchmarks and its enhanced robustness in perturbed environments.

However, our work also has some limitations. For example, current SCAS primarily focuses on continuous control tasks. In discrete settings, algorithmic components like state perturbation strategy would be different, which would be an interesting direction for future work. Moreover, we anticipate employing more advanced dynamics models, such as ensembles [73] and diffusion models [21], to further improve the performance of our method.

## Acknowledgment

We thank the anonymous reviewers for feedback on an early version of this paper. This work was supported by the National Key R&D Program of China under Grant 2018AAA0102801, National Natural Science Foundation of China under Grant 61827804.

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

# A   Related Work

**Model-free offline RL.**   In offline RL, extrapolation error and overestimation caused by OOD actions pose significant challenges. Among model-free solutions, value regularization methods penalize the $Q$-values of OOD actions [31, 2, 28, 36, 3, 72, 40], while policy constraint approaches compel the trained policy to be close to the behavior policy, either explicitly via divergence penalties [69, 30, 10], implicitly by weighted behavior cloning [47, 45, 67, 39], or directly through specific parameterization of the policy [12, 15]. Relatively independently, in-sample learning methods formulate the Bellman target using only the actions in the dataset to avoid OOD actions [5, 29, 76, 71]. Recently, some works aim to learn the optimal policy within the support of the dataset (known as in-support or in-sample optimal policy) in a theoretically sound way and are less affected by the average quality of the dataset [39, 40, 68]. However, existing popular offline RL approaches primarily focus on the OOD action issue during training and often neglect the OOD state issue during the test phase.

**Model-based offline RL.**   Model-based RL methods learn a model of the environment and generate synthetic data from that model to optimize the policy [60, 20, 24]. To ensure conservatism in offline RL, Kidambi et al. [25] and Yu et al. [73] estimate the uncertainty in the model and apply reward penalties for state-action pairs with high uncertainty. Some model-based approaches also introduce conservatism similarly to model-free ones, employing techniques like value regularization [74] and policy constraint [41]. Recently, Sun et al. [59] conducts uncertainty quantification through the inconsistency of Bellman estimations under the learned dynamics ensemble. However, model-based methods often come with a high computational burden [20], and their effectiveness relies heavily on the quality of the trained model [43]. In contrast, our algorithm leverages the dynamics model to propagate policy gradients, make one-step predictions, and regularize policy training, leading to significantly improved computational efficiency and relatively high prediction accuracy.

**OOD state correction.**   In offline RL, OOD state correction deserves more attention as the state deviation during the test phase can accumulate over time steps, severely degrading performance [34]. Existing limited solutions aim to train the policy to correct the agent from OOD states to ID states [75, 22]. Specifically, SDC [75] builds a dynamics model and a state transition model, and aligns the policy-induced next state distributions at OOD states with the state transition model. On the other hand, OSR [22] utilizes an inverse dynamics model to constrain the policy at OOD states. Compared with prior methods, our proposed SCAS efficiently unifies OOD state correction and OOD action suppression in offline RL and additionally achieves *value-aware* OOD state correction. The DICE series of works [44, 33, 38] share similar motivations with SCAS to some extent; however, there are significant differences between the two. Firstly, DICE is based on a linear programming framework of RL, while SCAS is based on a dynamic programming framework. Therefore, the theoretical foundations and learning paradigms of the two are inherently different. Secondly, SCAS only corrects encountered OOD states, whereas DICE algorithms require the policy-induced occupancy distribution to align with the dataset distribution. Therefore, DICE's constraints are stricter, potentially making it more susceptible to the average quality of the dataset. Lastly, theoretical and empirical evidence indicate that DICE algorithms have a problem of gradient cancellation [38], which imposes certain limitations on their practical effectiveness.

# B   Proofs

In this section, we present the proofs for the theories in the paper.

## B.1   Derivation of the Value-aware State Transition Distribution

We show that Eq. (6) is the optimal solution of the optimization problem Eq. (5):

$$\max_{N^*} \mathbb{E}_{s\sim\mathcal{D}} \left[ \alpha \mathbb{E}_{s'\sim N^*(\cdot|s)} V(s') - \mathrm{D_{KL}}(N^*(\cdot|s)\|N(\cdot|s)) \right] \tag{21}$$

We can optimize $N^*$ at each $s \in \mathcal{D}$ separately. Thus we consider the following optimization problem:

$$\max_{\tilde{N}} \alpha \mathop{\mathbb{E}}_{s' \sim \tilde{N}(\cdot|s)} V(s') - \mathrm{D}_{\mathrm{KL}}(\tilde{N}(\cdot|s) \| N(\cdot|s)) \tag{22}$$
$$s.t. \sum_{s'} \tilde{N}(s'|s) = 1, \ \forall s \in \mathcal{D}$$

This constrained optimization problem is convex, and the Lagrangian is:

$$\mathcal{L}(\tilde{N}) = \alpha \mathop{\mathbb{E}}_{s' \sim \tilde{N}(\cdot|s)} V(s') - \mathrm{D}_{\mathrm{KL}}(\tilde{N}(\cdot|s) \| N(\cdot|s)) + \nu \left( \sum_{s'} \tilde{N}(s'|s) - 1 \right) \tag{23}$$

The KKT condition gives:

$$\frac{\partial \mathcal{L}}{\partial \tilde{N}(s'|s)} = \alpha V(s') - \log \tilde{N}(s'|s) - 1 + \log N(s'|s) + \nu = 0 \tag{24}$$

Solving for $\tilde{N}$ gives the closed form solution $N^*$:

$$N^*(s'|s) = \exp\left(\alpha V(s') - 1 + \nu\right) N(s'|s), \ \forall s \sim \mathcal{D} \tag{25}$$

By the condition $\sum_{s'} N^*(s'|s) = 1$, we can directly solve the Lagrangian multiplier $\nu$ and replace $\exp(\nu - 1)$ with a normalization factor:

$$N^*(s'|s) = \frac{1}{Z(s)} \exp\left(\alpha V(s')\right) N(s'|s), \ \forall s \sim \mathcal{D} \tag{26}$$

where $Z(s) = \sum_{s'} \exp\left(\alpha V(s')\right) N(s'|s)$ is the normalization factor.

## B.2 Proof of Proposition 1

**Proposition 3** (Proposition 1 in the main paper). *Suppose that the environment dynamics is deterministic, then both $\bar{R}(\pi)$ and $\bar{R}_1(\pi)$ achieve their global maximum at the policy $\pi^*$, where[4]*

$$\pi^*(a|s) = \frac{1}{Z(s)} \exp\left(\alpha V(M(s,a))\right) \beta(a|s) \tag{27}$$

*The support of $\pi^*$ is within that of the behavior policy $\beta$:*

$$\mathrm{supp}(\pi^*(\cdot|s)) \subseteq \mathrm{supp}(\beta(\cdot|s)), \ \forall s \sim \mathcal{D} \tag{28}$$

*and $\pi^*$ makes the following equation hold:*

$$N^*(\cdot|s) = M(\cdot|s, \pi^*(\cdot|s)), \ \forall s \sim \mathcal{D} \tag{29}$$

*Proof.* We start with $\bar{R}(\pi)$.

$$\mathop{\mathrm{argmax}}_{\pi} \ \bar{R}(\pi) \tag{30}$$

$$= \mathop{\mathrm{argmax}}_{\pi} \mathop{\mathbb{E}}_{(s,s') \sim \mathcal{D}} \left[ \frac{1}{Z(s)} \exp\left(\alpha V(s')\right) \log M(s'|s, \pi(\cdot|s)) \right] \tag{31}$$

$$= \mathop{\mathrm{argmax}}_{\pi} \mathop{\mathbb{E}}_{s \sim \mathcal{D}} \mathop{\mathbb{E}}_{s' \sim N(\cdot|s)} \left[ \frac{1}{Z(s)} \exp\left(\alpha V(s')\right) \log M(s'|s, \pi(\cdot|s)) \right] \tag{32}$$

$$= \mathop{\mathrm{argmax}}_{\pi} \mathop{\mathbb{E}}_{s \sim \mathcal{D}} \mathop{\mathbb{E}}_{s' \sim N^*(\cdot|s)} \left[ \log M(s'|s, \pi(\cdot|s)) \right] \tag{33}$$

$$= \mathop{\mathrm{argmin}}_{\pi} \mathop{\mathbb{E}}_{s \sim \mathcal{D}} \mathop{\mathbb{E}}_{s' \sim N^*(\cdot|s)} \left[ \log N^*(s'|s) - \log M(s'|s, \pi(\cdot|s)) \right] \tag{34}$$

$$= \mathop{\mathrm{argmin}}_{\pi} \mathop{\mathbb{E}}_{s \sim \mathcal{D}} \mathrm{D}_{\mathrm{KL}}(N^*(\cdot|s) \| M(\cdot|s, \pi(\cdot|s))) \tag{35}$$

---

[4]Here for clarity, we use the notation $M$ with slightly different meanings in different cases: in the stochastic setting, $M : \mathcal{S} \times \mathcal{A} \to \Delta(\mathcal{S})$; in the deterministic setting, $M : \mathcal{S} \times \mathcal{A} \to \mathcal{S}$.

The third equality holds because of the relationship between $N^*$ and $N$ in Eq. (6):

$$N^*(s'|s) = \frac{1}{Z(s)} \exp\left(\alpha V\left(s'\right)\right) N(s'|s), \ \forall s \sim \mathcal{D} \tag{36}$$

Therefore, the maximizer of $\bar{R}(\pi)$ is equal to the solution of the minimization problem in Eq. (35). Now consider the two distributions $N^*(\cdot|s)$ and $M(\cdot|s, \pi(\cdot|s))$ in Eq. (35).

$$N^*(s'|s) = \frac{1}{Z(s)} \exp\left(\alpha V\left(s'\right)\right) N(s'|s) \tag{37}$$

$$= \frac{1}{Z(s)} \exp\left(\alpha V\left(s'\right)\right) \sum_a \beta(a|s) M(s'|s, a) \tag{38}$$

For analytical clarity, we use the notation $M$ with slightly different meanings in different cases: in the stochastic setting, $M : \mathcal{S} \times \mathcal{A} \to \Delta(\mathcal{S})$; in the deterministic setting, $M : \mathcal{S} \times \mathcal{A} \to \mathcal{S}$. With the deterministic dynamics assumption,

$$N^*(s'|s) = \frac{1}{Z(s)} \exp\left(\alpha V\left(s'\right)\right) \sum_a \beta(a|s) \mathbb{I}\left[M(s, a) = s'\right] \tag{39}$$

$$= \sum_a \frac{1}{Z(s)} \exp\left(\alpha V\left(s'\right)\right) \beta(a|s) \mathbb{I}\left[M(s, a) = s'\right] \tag{40}$$

$$= \sum_a \frac{1}{Z(s)} \exp\left(\alpha V\left(M(s, a)\right)\right) \beta(a|s) \mathbb{I}\left[M(s, a) = s'\right] \tag{41}$$

On the other hand,

$$M(s'|s, \pi(\cdot|s)) = \sum_a M(s'|s, a)\pi(a|s) \tag{42}$$

$$= \sum_a \pi(a|s) \mathbb{I}\left[M(s, a) = s'\right] \tag{43}$$

Now we define $\pi^*(a|s)$ as

$$\pi^*(a|s) := \frac{1}{Z(s)} \exp\left(\alpha V\left(M(s, a)\right)\right) \beta(a|s) \tag{44}$$

We first show that $\pi^*$ is a valid policy, that is, $\pi^*$ is normalized.

$$\pi^*(a|s) = \frac{1}{Z(s)} \exp\left(\alpha V\left(M(s, a)\right)\right) \beta(a|s) \tag{45}$$

$$= \frac{\exp\left(\alpha V\left(M(s, a)\right)\right) \beta(a|s)}{\sum_{s'} \exp\left(\alpha V\left(s'\right)\right) N(s'|s)} \tag{46}$$

$$= \frac{\exp\left(\alpha V\left(M(s, a)\right)\right) \beta(a|s)}{\sum_{s'} \exp\left(\alpha V\left(s'\right)\right) \sum_a \beta(a|s) M(s'|s, a)} \tag{47}$$

$$= \frac{\exp\left(\alpha V\left(M(s, a)\right)\right) \beta(a|s)}{\sum_a \sum_{s'} \exp\left(\alpha V\left(s'\right)\right) \beta(a|s) M(s'|s, a)} \tag{48}$$

$$= \frac{\exp\left(\alpha V\left(M(s, a)\right)\right) \beta(a|s)}{\sum_a \exp\left(\alpha V\left(M(s, a)\right)\right) \beta(a|s)} \tag{49}$$

Therefore, $\sum_a \pi^*(a|s) = 1$.

Substitute Eq. (44) into Eq. (41),

$$N^*(s'|s) = \sum_a \pi^*(a|s) \mathbb{I}\left[M(s, a) = s'\right] \tag{50}$$

Comparing Eq. (43) with Eq. (50), it holds that $N^*(s'|s) = M(s'|s, \pi^*(\cdot|s)), \forall s \sim \mathcal{D}$. As a result,

$$\mathbb{E}_{s \sim \mathcal{D}} D_{\mathrm{KL}}(N^*(\cdot|s) \| M(\cdot|s, \pi^*(\cdot|s))) = 0 \tag{51}$$

Considering the non-negativity of KL divergence, the optimization problem in Eq. (35) achieves its global minimum at $\pi^*$. Therefore, $\bar{R}(\pi)$ also achieves its global maximum at $\pi^*$.

Now we consider $\bar{R}_1(\pi)$.

$$\underset{\pi}{\operatorname{argmax}} \ \bar{R}_1(\pi) \tag{52}$$

$$= \underset{\pi}{\operatorname{argmax}} \ \underset{(s,s')\sim\mathcal{D}}{\mathbb{E}} \left[ \exp\left(\alpha\left(V\left(s'\right) - V\left(s\right)\right)\right) \log M(s'|s, \pi(\cdot|s)) \right] \tag{53}$$

$$= \underset{\pi}{\operatorname{argmax}} \ \underset{(s,s')\sim\mathcal{D}}{\mathbb{E}} \left[ \frac{Z(s)}{\exp\left(\alpha V\left(s\right)\right) Z(s)} \exp\left(\alpha V\left(s'\right)\right) \log M(s'|s, \pi(\cdot|s)) \right] \tag{54}$$

$$= \underset{\pi}{\operatorname{argmax}} \ \underset{s\sim\mathcal{D}}{\mathbb{E}} \ \underset{s'\sim N(\cdot|s)}{\mathbb{E}} \left[ \frac{Z(s)}{\exp\left(\alpha V\left(s\right)\right) Z(s)} \exp\left(\alpha V\left(s'\right)\right) \log M(s'|s, \pi(\cdot|s)) \right] \tag{55}$$

$$= \underset{\pi}{\operatorname{argmax}} \ \underset{s\sim\mathcal{D}}{\mathbb{E}} \ \underset{s'\sim N^*(\cdot|s)}{\mathbb{E}} \left[ \frac{Z(s)}{\exp\left(\alpha V\left(s\right)\right)} \log M(s'|s, \pi(\cdot|s)) \right] \tag{56}$$

$$= \underset{\pi}{\operatorname{argmin}} \ \underset{s\sim\mathcal{D}}{\mathbb{E}} \ \underset{s'\sim N^*(\cdot|s)}{\mathbb{E}} \left[ \frac{Z(s)}{\exp\left(\alpha V\left(s\right)\right)} \left(\log N^*(s'|s) - \log M(s'|s, \pi(\cdot|s))\right) \right] \tag{57}$$

$$= \underset{\pi}{\operatorname{argmin}} \ \underset{s\sim\mathcal{D}}{\mathbb{E}} \left[ \frac{Z(s)}{\exp\left(\alpha V\left(s\right)\right)} \mathrm{D}_{\mathrm{KL}}(N^*(\cdot|s)\|M(\cdot|s, \pi(\cdot|s))) \right] \tag{58}$$

The fourth equality holds because of the relationship between $N^*$ and $N$ in Eq. (36).

As shown above, it holds that $N^*(s'|s) = M(s'|s, \pi^*(\cdot|s)), \forall s \sim \mathcal{D}$. As a result,

$$\underset{s\sim\mathcal{D}}{\mathbb{E}} \left[ \frac{Z(s)}{\exp\left(\alpha V\left(s\right)\right)} \mathrm{D}_{\mathrm{KL}}(N^*(\cdot|s)\|M(\cdot|s, \pi(\cdot|s))) \right] = 0 \tag{59}$$

Considering $Z(s)/\exp\left(\alpha V\left(s\right)\right) > 0$ and the non-negativity of KL divergence, the optimization problem in Eq. (58) achieves its global minimum at $\pi^*$. Therefore, $\bar{R}_1(\pi)$ also achieves its global maximum at $\pi^*$.

In conclusion, when the environment dynamics is deterministic, both $\bar{R}(\pi)$ and $\bar{R}_1(\pi)$ achieve their global maximum at the policy $\pi^*$, and $\pi^*$ makes the following equation hold:

$$N^*(\cdot|s) = M(\cdot|s, \pi^*(\cdot|s)), \ \forall s \sim \mathcal{D} \tag{60}$$

Moreover, because $\pi^*(a|s) = \frac{1}{Z(s)} \exp\left(\alpha V\left(M(s,a)\right)\right) \beta(a|s)$, the support of $\pi^*$ is included by that of the behavior policy $\beta$:

$$\operatorname{supp}(\pi^*(\cdot|s)) \subseteq \operatorname{supp}(\beta(\cdot|s)), \ \forall s \sim \mathcal{D} \tag{61}$$

$\square$

### B.3   Proof of Proposition 2

**Proposition 4** (Proposition 2 in the main paper). *When the dynamics is stochastic, the maximizers of both $\bar{R}(\pi)$ and $\bar{R}_1(\pi)$ are constrained within the support of the behavior policy:*

$$\operatorname{supp}(\pi^*(\cdot|s)) \subseteq \operatorname{supp}(\beta(\cdot|s)), \ \forall s \sim \mathcal{D} \tag{62}$$

$$\operatorname{supp}(\pi_1^*(\cdot|s)) \subseteq \operatorname{supp}(\beta(\cdot|s)), \ \forall s \sim \mathcal{D} \tag{63}$$

*Proof.* We start with $\bar{R}(\pi)$.

$$\bar{R}(\pi) := \underset{(s,s')\sim\mathcal{D}}{\mathbb{E}} \left[ \frac{1}{Z(s)} \exp\left(\alpha V\left(s'\right)\right) \log M(s'|s, \pi(\cdot|s)) \right] \tag{64}$$

$$= \underset{(s,s')\sim\mathcal{D}}{\mathbb{E}} \left[ \frac{1}{Z(s)} \exp\left(\alpha V\left(s'\right)\right) \log\left(\sum_a M(s'|s, a)\pi(a|s)\right) \right] \tag{65}$$

Let $\pi$ denote any valid policy. For $\forall s \in \mathcal{D}$, define $\epsilon(s)$ and $n(s)$ as follows:

$$\epsilon(s) := \sum_a \mathbb{I}[\beta(a|s) = 0]\pi(a|s) \tag{66}$$

$$n(s) := \sum_a \mathbb{I}[\beta(a|s) > 0] \tag{67}$$

For $\forall s \in \mathcal{D}$, there exists at least one action $a$ such that $(s, a) \in \mathcal{D}$. Thus it holds that $n(s) > 0, \forall s \in \mathcal{D}$. Then for $\forall s \in \mathcal{D}, \forall \pi$, define $\pi_{\text{in}}$ as follows:

$$\pi_{\text{in}}(a|s) = \begin{cases} \pi(a|s) + \frac{\epsilon(s)}{n(s)}, & \beta(a|s) > 0, \\ 0, & \beta(a|s) = 0. \end{cases} \tag{68}$$

$\pi_{\text{in}}$ can be seen as a projection of $\pi$ onto $\beta$'s support. Besides, for $\forall s \in \mathcal{D}$,

$$\sum_a \pi_{\text{in}}(a|s) = \sum_a \mathbb{I}[\beta(a|s) > 0] \left( \pi(a|s) + \frac{\epsilon(s)}{n(s)} \right) \tag{69}$$

$$= \sum_a \mathbb{I}[\beta(a|s) > 0]\pi(a|s) + \epsilon(s) \tag{70}$$

$$= \sum_a \mathbb{I}[\beta(a|s) > 0]\pi(a|s) + \sum_a \mathbb{I}[\beta(a|s) = 0]\pi(a|s) \tag{71}$$

$$= \sum_a \pi(a|s) \tag{72}$$

$$= 1 \tag{73}$$

Thus $\pi_{\text{in}}$ is a valid policy.

Now we compare $\bar{R}(\pi_{\text{in}})$ with $\bar{R}(\pi)$. For $\forall (s, s') \in \mathcal{D}$,

$$\sum_a M(s'|s, a)\pi_{\text{in}}(a|s) - \sum_a M(s'|s, a)\pi(a|s) \tag{74}$$

$$= \sum_a M(s'|s, a) \left( \pi_{\text{in}}(a|s) - \pi(a|s) \right) \tag{75}$$

$$= \sum_{\{a|\beta(a|s)>0\}} M(s'|s, a) \left( \pi_{\text{in}}(a|s) - \pi(a|s) \right) \tag{76}$$

$$= \sum_{\{a|\beta(a|s)>0\}} M(s'|s, a)\frac{\epsilon(s)}{n(s)} \tag{77}$$

$$\geq \quad 0 \tag{78}$$

The second equality holds because, in tabular MDPs, the empirical dynamics model $M$ exactly computes the conditional distribution observed in the dataset. For transitions not contained in the dataset, $M = 0$ [12]. The final inequality holds because $\epsilon(s) \geq 0$.

Therefore,

$$\bar{R}(\pi_{\text{in}}) - \bar{R}(\pi) \tag{79}$$

$$= \mathop{\mathbb{E}}_{(s,s')\sim\mathcal{D}} \left[ \frac{1}{Z(s)} \exp\left( \alpha V\left( s' \right) \right) \log\left( \frac{\sum_a M(s'|s, a)\pi_{\text{in}}(a|s)}{\sum_a M(s'|s, a)\pi(a|s)} \right) \right] \tag{80}$$

$$\geq \mathop{\mathbb{E}}_{(s,s')\sim\mathcal{D}} \left[ \frac{1}{Z(s)} \exp\left( \alpha V\left( s' \right) \right) \log\left( 1 \right) \right] \tag{81}$$

$$\geq \quad 0 \tag{82}$$

Now suppose $\pi$ is not constrained within the support of the behavior policy at some state $s_1 \in \mathcal{D}$: $\text{supp}(\pi(\cdot|s_1)) \not\subseteq \text{supp}(\beta(\cdot|s_1))$. That is, $\exists \tilde{a}_1$ such that $\pi(\tilde{a}_1|s_1) > 0$ and $\beta(\tilde{a}_1|s_1) = 0$. Thus it holds that $\epsilon(s_1) = \sum_a \mathbb{I}[\beta(a|s_1) = 0]\pi(a|s_1) > 0$. On the other hand, since $s_1 \in \mathcal{D}$, there exists at least one action $a_1$ and one state $s_1'$ such that $(s_1, a_1, s_1') \in \mathcal{D}$. Thus it holds that $\beta(a_1|s_1) > 0$ and $M(s_1'|s_1, a_1) > 0$. As a result,

$$\sum_a M(s_1'|s_1, a)\pi_{\text{in}}(a|s_1) - \sum_a M(s_1'|s_1, a)\pi(a|s_1) \tag{83}$$

$$= \sum_{\{a|\beta(a|s_1)>0\}} M(s_1'|s_1, a)\frac{\epsilon(s_1)}{n(s_1)} \tag{84}$$

$$> \quad 0 \tag{85}$$

In such case, $\bar{R}(\pi_{\text{in}}) > \bar{R}(\pi)$. Therefore, if $\pi$ is not constrained within the support of the behavior policy at some state $s_1 \in \mathcal{D}$, we can find another policy $\pi_{\text{in}}$ that is constrained within the support of the behavior policy and achieves higher objective function $\bar{R}(\pi_{\text{in}})$. Consequently, $\bar{R}(\pi)$ must achieve its maximum at support constrained policy $\pi^*$: $\text{supp}(\pi^*(\cdot|s)) \subseteq \text{supp}(\beta(\cdot|s))$, $\forall s \sim \mathcal{D}$.

Now we consider $\bar{R}_1(\pi)$.

$$\bar{R}_1(\pi) := \mathop{\mathbb{E}}_{(s,s')\sim\mathcal{D}} \left[ \exp\left( \alpha \left( V\left( s' \right) - V\left( s \right) \right) \right) \log M(s'|s, \pi(\cdot|s)) \right] \tag{86}$$

$$= \mathop{\mathbb{E}}_{(s,s')\sim\mathcal{D}} \left[ \exp\left( \alpha \left( V\left( s' \right) - V\left( s \right) \right) \right) \log\left( \sum_a M(s'|s, a)\pi(a|s) \right) \right] \tag{87}$$

With the same definition of $\epsilon(s)$, $n(s)$ and $\pi_{\text{in}}$ as in Eq. (66), Eq. (67) and Eq. (68), it also holds that

$$\bar{R}_1(\pi_{\text{in}}) - \bar{R}_1(\pi) \tag{88}$$

$$= \mathop{\mathbb{E}}_{(s,s')\sim\mathcal{D}} \left[ \exp\left( \alpha \left( V\left( s' \right) - V\left( s \right) \right) \right) \log\left( \frac{\sum_a M(s'|s, a)\pi_{\text{in}}(a|s)}{\sum_a M(s'|s, a)\pi(a|s)} \right) \right] \tag{89}$$

$$\geq \mathop{\mathbb{E}}_{(s,s')\sim\mathcal{D}} \left[ \exp\left( \alpha \left( V\left( s' \right) - V\left( s \right) \right) \right) \log\left( 1 \right) \right] \tag{90}$$

$$\geq \quad 0 \tag{91}$$

As before, when supposing $\pi$ is not constrained within the support of the behavior policy at some state $s_1 \in \mathcal{D}$, it holds that $\bar{R}_1(\pi_{\text{in}}) > \bar{R}_1(\pi)$. Therefore, $\bar{R}_1(\pi)$ must achieve its maximum at support constrained policy $\pi_1^*$: $\text{supp}(\pi_1^*(\cdot|s)) \subseteq \text{supp}(\beta(\cdot|s))$, $\forall s \sim \mathcal{D}$.

In conclusion, when the environment dynamics is stochastic, the maximizers of both $\bar{R}(\pi)$ and $\bar{R}_1(\pi)$ are constrained within the support of the behavior policy:

$$\text{supp}(\pi^*(\cdot|s)) \subseteq \text{supp}(\beta(\cdot|s)), \ \text{supp}(\pi_1^*(\cdot|s)) \subseteq \text{supp}(\beta(\cdot|s)), \ \forall s \sim \mathcal{D} \tag{92}$$

$\square$

## C   Further Discussions

### C.1   Rationale for Choosing $\exp(\alpha V(s))$ as the Empirical Normalizer

Firstly, choosing $\exp(\alpha V(s))$ is intended to obtain something similar to the advantage function. With this normalizer, the weight of our regularizer is $\exp(\alpha(V(s') - V(s)))$, which is comparable to the weight $\exp(\alpha A(s, a))$ in Advantage Weighted Regression (AWR) [47]. Here, $V(s') - V(s)$ represents the relative advantage of the next state $s'$ compared to the current state $s$, while $A(s, a)$ reflects the relative advantage of taking action $a$ in $s$ compared to following the current policy. Comparison of the objectives of SCAS and AWR:

$$\text{SCAS:} \quad \exp(\alpha(V(s') - V(s))) \log(M(s'|\hat{s}, \pi(\hat{s}))) \tag{93}$$

$$\text{AWR:} \quad \exp(\alpha A(s, a)) \log \pi(a|s) \tag{94}$$

Secondly, as discussed in the paper, introducing any normalizer that depends only on $s$ (independent of $s'$) does not affect the development and analysis of our method; it is merely for computational stability. In AWR-based methods, there also exists a normalizer $Z(s)$ and they usually disregard it [47, 45]. The rationale behind this is similar.

### C.2   Pessimism and Robustness in SCAS

In a specific sense, SCAS, which unifies OOD state correction and OOD action suppression, also integrates pessimism and state robustness. (1) Regarding pessimism: The OOD action suppression effect of SCAS aligns with the pessimism commonly discussed in offline RL work (being pessimistic about OOD actions) [31, 70, 30, 3, 54]. Unlike traditional policy constraint methods [69, 30, 10, 47], our approach does not require the training policy to align with the behavior policy; it only requires the successor states to be within the dataset support, which is a more relaxed constraint. (2) Regarding state robustness: The OOD state correction effect of SCAS is aimed at improving the agent's

robustness to OOD states during the test phase. Compared with previous works, SCAS unifies OOD state correction and OOD action suppression and additionally achieves value-aware OOD state correction. Some offline RL literature on state robustness differs from our approach; they typically consider noisy observations [72], such as sensor errors. In contrast, SCAS addresses state robustness concerning actual OOD states encountered during test time, rather than noisy observations.

## C.3 Regularization Effect at ID States

In SCAS, there is regularization on the policy's output actions at ID states. In our regularizer, the perturbed states $\hat{s}$ are sampled from $\mathcal{N}(s, \sigma^2)$, and a large portion of $\hat{s}$ will fall near the original ID state $s$ or even be approximately equal to $s$. Therefore, the policy's output actions at ID states are also regularized. For this part of the regularization, its role is equivalent to the ID state regularizer analyzed in Section 4, which has been theoretically shown to have the effect of suppressing OOD actions. Moreover, the experimental results in Section 6 also demonstrate that our OOD state correction regularizer addresses the traditional issue with OOD actions.

## C.4 Differences between the OOD Action Issue and the OOD State Issue

We further elucidate the differences between the well-known OOD action issue and the OOD state issue we analyzed. Most offline RL works focus on the OOD action issue in the training phase. That is, the trained policy outputs OOD actions to compute the target Q, which results in extrapolation error and value divergence during training [12]. In contrast, the OOD state issue we defined and analyzed is in the test phase. That is, the agent can enter states out of the offline dataset during test, potentially resulting in catastrophic failure.

## D  Experimental Details

Table 2: Hyperparameters in SCAS.

|  | Hyperparameter | Value |
|---|---|---|
| Policy training | Optimizer | Adam [26] |
|  | Critic learning rate | $3 \times 10^{-4}$ |
|  | Actor learning rate | $2 \times 10^{-4}$ with cosine schedule |
|  | Batch size | 256 |
|  | Discount factor | 0.99 |
|  | Gradient Steps | $10^6$ |
|  | Target network update rate | 0.005 |
|  | Policy update frequency | 2 |
|  | Number of Critics | 4 |
|  | Inverse temperature $\alpha$ | 5 |
|  | Balance coefficient $\lambda$ | 0.25 |
|  | Noise scale $\sigma$ | 0.003 |
| Dynamics training | Optimizer | Adam |
|  | Learning rate | $1 \times 10^{-3}$ |
|  | Batch size | 256 |
|  | Gradient Steps | $5 \times 10^5$ |
| Architecture | Actor | input-256-256-output |
|  | Critic | input-256-256-1 |
|  | Dynamics | input-256-256-256-256-output |

All hyperparameters of SCAS are included in Table 2. Note that we use this same set of hyperparameters to obtain all the results reported in this paper (except for parameter study). Following TD3+BC [10], we normalize the states in all datasets except for antmaze-large. We clip the exponentiated weight $\exp\left(\alpha V_\theta\left(s'\right) - \alpha V_\theta\left(s\right)\right)$ in Eq. (19) to $(-\infty, 50]$. Following the suggestions in the benchmark [9], we subtract 1 from the rewards for the Antmaze datasets.

Our evaluation criteria follow those used in most previous works. For the Gym locomotion tasks, we average returns over 10 evaluation trajectories and 5 random seeds, while for the Ant Maze tasks, we average over 100 evaluation trajectories and 5 random seeds. The reported results are the normalized scores, which are offered by the D4RL benchmark [9] to measure how the learned policy compared with random and expert policy:

$$\text{D4RL score} = 100 \times \frac{\text{learned policy return} - \text{random policy return}}{\text{expert policy return} - \text{random policy return}}$$

The results of baselines reported in Table 1 are obtained as follows. We re-run OSR [22] on all datasets using their official codebase[5] and tune the hyperparameters for each dataset as specified in their paper. We implement SDC [75] and re-run it on all datasets. We use the SDC-related hyperparameters as specified in their paper, and sweep the CQL-related hyperparameters in {1,2,5,10,20} for each dataset. We re-run OneStep RL [5] on all datasets using their official codebase[6] and the default hyperparameters. We implement BC [48] based on the TD3+BC repository[7] and re-run it on all datasets. The results of other baselines are taken from [3] and [68]. The runtime in Table 1 is obtained by running offline RL algorithms on halfcheetah-medium-replay-v2 on a GeForce RTX 3090.

Figures 1(a) to 1(d) share the same embedding function obtained by running t-SNE on the set of all 200,000 samples (50,000 samples each from the dataset, CQL, TD3+BC, and SCAS). This ensures a clear visual comparison. Figure 1(d) contains all the 200,000 samples, which is the union of the points in Figures 1(a) to 1(c).

# E    Additional Experimental Results

## E.1    Additional Value Estimation Results

Under the same setting of Figure 2, we conduct experiments on the additional datasets. The results are shown in Figure 5. We omit the Q values of Off-policy RL, SDC w/o CQL, and OSR w/o CQL at higher numbers of optimization steps, because these Q values diverge in the early learning stage, and plotting their Q values at later optimization steps would result in an excessive range on the vertical axis. The additional results also show that only SCAS's OOD state correction term can achieve OOD action suppression and prevent value over-estimation.

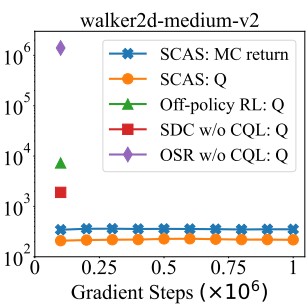 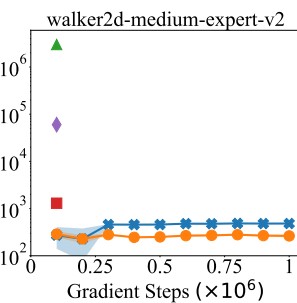

Figure 5: Oracle Q-values of SCAS (estimated by MC return) and learned Q-values of SCAS and other algorithms across optimization steps. Here Off-policy RL is SCAS with weight $\lambda = 0$ in Eq. (20). Only SCAS's OOD state correction term can achieve OOD action suppression and prevent value over-estimation (divergence).

## E.2    Additional Results on OOD State Correction

To further examine the OOD state correction effects of SCAS, we conduct experiments on a modified D4RL maze2d-open-v0 [9]. It is a 2D point robot navigation task in a rectangle map with vertices $(0,0)$ and $(3,5)$. The agent needs to reach the goal at $(2,3)$. We modify the dataset by removing all

---

[5]https://github.com/Jack10843/OSR
[6]https://github.com/davidbrandfonbrener/onestep-rl
[7]https://github.com/sfujim/TD3_BC

the transitions containing states in a rectangle with vertices $(0, 0)$ and $(1.5, 2.5)$. During test, we let the initial state be randomly distributed in this OOD region. We train algorithms over $10^6$ gradient steps and average returns over 1000 evaluation trajectories.

The results of BC [48], TD3+BC [10], CQL [31], MOPO [73], IQL [29], and SCAS are reported in Table 3. With the OOD state correction signals, SCAS corrects the agent out of the OOD region more quickly and stably, achieving significantly better performance than typical offline RL methods.

Table 3: Comparisons in modified maze2d-open-v0 over five random seeds.

|  | BC | TD3+BC | CQL | MOPO | IQL | SCAS |
|---|---|---|---|---|---|---|
| Steps out of OOD | 84.7±44.7 | 58.0±35.7 | 63.8±33.0 | 50.6±25.4 | 37.7±6.7 | **22.8±3.1** |
| D4RL score | 38.5±25.4 | 63.9±39.3 | 41.2±42.0 | 110.1±78.8 | 335.0±114.9 | **571.9±2.7** |

### E.3 Comparisons on the NeoRL Benchmark

Table 4: Averaged normalized scores on the NeoRL benchmark over four random seeds.

|  | BC | TD3BC | CQL | EDAC | MOPO | MOBILE | SCAS |
|---|---|---|---|---|---|---|---|
| Hopper-High | 43.1 | 75.3 | 76.6 | 52.5 | 11.5 | 87.8 | **100.5±7.8** |
| Hopper-Med | 51.3 | 70.3 | 64.5 | 44.9 | 1.0 | 51.1 | **94.6±9.3** |
| Hopper-Low | 15.1 | 15.8 | 16.0 | 18.3 | 6.2 | 17.4 | **19.7±1.2** |
| Walker2d-High | 72.6 | 69.6 | **75.3** | **75.5** | 18.0 | **74.9** | **74.6±0.7** |
| Walker2d-Med | 48.7 | 58.5 | 57.3 | 57.6 | 39.9 | 62.2 | **63.4±1.0** |
| Walker2d-Low | 28.5 | 43.0 | **44.7** | 40.2 | 11.6 | 37.6 | 34.4±1.3 |
| HalfCheetah-High | 71.3 | 75.3 | 77.4 | 81.4 | 65.9 | **83.0** | 77.0±0.5 |
| HalfCheetah-Med | 49.0 | 52.3 | 54.6 | 54.9 | 62.3 | **77.8** | 53.1±0.1 |
| HalfCheetah-Low | 29.1 | 30.0 | 38.2 | 31.3 | 40.1 | **54.7** | 31.5±0.2 |
| total | 408.7 | 490.1 | 504.6 | 456.6 | 256.5 | **546.5** | **548.7** |
| hyperparameter tuning | **w/o** | w/ | w/ | w/ | w/ | w/ | **w/o** |

We also evaluate SCAS on the NeoRL benchmark [49]. NeoRL is a benchmark designed to simulate real-world scenarios by collecting datasets using a more conservative policy, aligning closely with realistic data collection scenarios. The narrow and limited data makes it challenging for offline RL algorithms. The results are shown in Table 4. The results of baselines are taken directly from the MOBILE paper [59]. According to Appendix C in [59], these results are obtained by tuning hyperparameters per dataset. For SCAS, we use the same fixed set of hyperparameters as specified in Appendix D. Without additional hyperparameter tuning, SCAS still performs comparably to MOBILE and outperforms other baselines in total scores.

### E.4 Comparisons with Additional Baselines

The original SCAS requires only one single hyperparameter configuration in implementations. For a fair comparison with DW [19], EDAC [2], RORL [72], SQL [71], and EQL [71], we roughly select $\lambda$ from {0.025, 0.25} for each dataset, referring to this variant as SCAS-ht. The results of SCAS-ht and these methods are reported in Table 5. Among the ensemble-free methods, SCAS-ht achieves the highest performance in both mujoco locomotion and antmaze domains. Compared with ensemble-based methods, SCAS-ht also performs better on antmaze tasks. DW [19] reweights ID data points by their values for behavior regularization and does not account for OOD states during the test phase. In contrast, our approach considers an OOD state correction scenario, resulting in enhanced robustness during the test phase and better performance.

### E.5 Results of Combining SCAS Regularizer into Various Offline RL Objectives

The SCAS regularizer is compatible with various offline RL objectives. We conduct experiments to combine SCAS with CQL [31], IQL [29], and TD3BC [10]. Comparisons between these combined

Table 5: Comparisons with additional baselines on the D4RL benchmark. Here SCAS-ht means SCAS with slight hyperparameter tuning, selecting $\lambda$ from $\{0.025, 0.25\}$. The results of SCAS-ht are averaged over 5 random seeds and the others are taken from their papers.

| Dataset | Ensemble-free | | | | | | Ensemble-based | |
|---|---|---|---|---|---|---|---|---|
| | DW+CQL | DW+IQL | SQL | EQL | DQL | SCAS-ht | EDAC | RORL |
| halfcheetah-med | 46.5 | 47.7 | 48.3 | 47.2 | 51.1 | **58.5**±1.1 | 65.9 | **66.8** |
| hopper-med | 66.1 | 62.5 | 75.5 | 74.6 | 90.5 | **102.5**±0.3 | 101.6 | **104.8** |
| walker2d-med | 82.1 | 80.8 | 84.2 | 83.2 | 87 | **90.8**±2.6 | 92.5 | **102.4** |
| halfcheetah-med-rep | 45.1 | 44.6 | 44.8 | 44.5 | 47.8 | **52.9**±1.4 | 61.3 | **61.9** |
| hopper-med-rep | 88.6 | 79.7 | 99.7 | 98.1 | 101.3 | **101.6**±1.0 | 101.0 | **102.8** |
| walker2d-med-rep | 75.3 | 65.1 | 81.2 | 76.6 | **95.5** | 88.1±4.2 | 87.1 | **90.4** |
| halfcheetah-med-exp | 86.1 | 93.7 | 94.0 | 90.6 | **96.8** | 91.7±2.7 | 106.3 | **107.8** |
| hopper-med-exp | 92.9 | 81.0 | **111.8** | 105.5 | 111.1 | 109.7±3.5 | 110.7 | **112.7** |
| walker2d-med-exp | 109.7 | 109.7 | 110.0 | 110.2 | 110.1 | **110.8**±1.0 | 114.7 | **121.2** |
| locomotion total | 692.4 | 664.8 | 749.5 | 730.5 | 791.2 | **806.6** | 841.1 | **870.8** |
| antmaze-umaze | 72.7 | 81.3 | 92.2 | 93.2 | **93.4** | 90.4±3.6 | 0.0 | **96.7** |
| antmaze-umaze-div | 34.0 | 61.0 | **74.0** | 65.4 | 66.2 | 66.4±14.3 | 0.0 | **90.7** |
| antmaze-med-play | 4.0 | 78.7 | 80.2 | 77.5 | 76.6 | **83.6**±3.1 | 0.0 | **76.3** |
| antmaze-med-div | 1.3 | 64.7 | 79.1 | 70.0 | 78.6 | **84.6**±5.0 | 0.0 | **69.3** |
| antmaze-large-play | 2.0 | 40.0 | 53.2 | 45.6 | 46.4 | **59.4**±5.0 | 0.0 | **16.3** |
| antmaze-large-div | 0.0 | 42.0 | 52.3 | 42.5 | **56.6** | 56.2±5.4 | 0.0 | **41.0** |
| antmaze total | 114.0 | 367.7 | 431.0 | 394.2 | 417.8 | **440.6** | 0.0 | **390.3** |

Table 6: Comparisons on the D4RL benchmark. Here +SCAS means adding the SCAS regularizer. The results are averaged over 5 random seeds.

| Dataset | CQL | CQL +SCAS | TD3BC | TD3BC +SCAS | IQL | IQL +SCAS | SCAS |
|---|---|---|---|---|---|---|---|
| halfcheetah-med | **47.0** | 46.5 | **48.3** | 44.1 | **47.4** | 46.8 | 46.6 |
| hopper-med | 53.0 | **96.1** | 59.3 | **66.6** | 66.2 | **76.8** | 102.5 |
| walker2d-med | 73.3 | **84.9** | **83.7** | 81.9 | 78.3 | **84.0** | 82.3 |
| halfcheetah-med-rep | **45.5** | 43.6 | **44.6** | 40.5 | **44.2** | **44.2** | 44.0 |
| hopper-med-rep | 88.7 | **100.2** | 60.9 | **79.4** | 94.7 | **102.3** | 101.6 |
| walker2d-med-rep | **81.8** | 78.6 | **81.8** | 76.2 | 73.8 | **76.2** | 78.1 |
| halfcheetah-med-exp | 75.6 | **92.9** | 90.7 | 89.6 | 86.7 | **92.7** | 91.7 |
| hopper-med-exp | 105.6 | **108.2** | 98.0 | **108.9** | 91.5 | **101.9** | 109.7 |
| walker2d-med-exp | **107.9** | 104.3 | **110.1** | 106.0 | **109.6** | 105.4 | 108.4 |
| total | 678.4 | **755.5** | 677.4 | **693.2** | 692.4 | **730.3** | **764.9** |

algorithms and the original ones are shown in Table 6. We find that applying the SCAS regularizer leads to improved performance for these popular algorithms, which could be attributed to the OOD state correction effects of SCAS. However, we also find that these combined methods do not achieve better performance than the original SCAS (comparable on most tasks and worse on some tasks). We hypothesize that this is because SCAS already has the effect of OOD action suppression, and when combined with offline RL objectives that also aim for OOD action suppression, it may become overly conservative. As a result, the combined algorithms may perform worse than the original SCAS on some sub-optimal datasets.

## E.6 Additional Parameter Study Results

In this section, we present additional parameter study results conducted on four challenging Antmaze tasks, including antmaze-large-play-v2, antmaze-large-diverse-v2, antmaze-medium-play-v2, and antmaze-medium-diverse-v2.

**Inverse Temperature** $\alpha$. The inverse temperature $\alpha$ is the key hyperparameter in SCAS for achieving value-aware OOD state correction. It controls the significance of the values of next states in SCAS's

OOD state correction. If $\alpha = 0$, the effect corresponds to vanilla OOD state correction. As $\alpha$ gets larger, SCAS is more inclined to correct the agent to the high-value ID states. Thus we can assess the effectiveness of value-aware OOD state correction compared to vanilla OOD state correction by varying $\alpha$. Here we test SCAS with different $\alpha$ and the results are shown in Figure 6. We observe that a large $\alpha$ is crucial for achieving good performance on all the antmaze tasks, clearly demonstrating the effectiveness of our *value-aware* OOD state correction. However, too large $\alpha$ ($\alpha = 10$) induces less satisfying performance, probably due to the increased variance of the learning objective. In general, we find that choosing $\alpha = 5$ leads to the best performance.

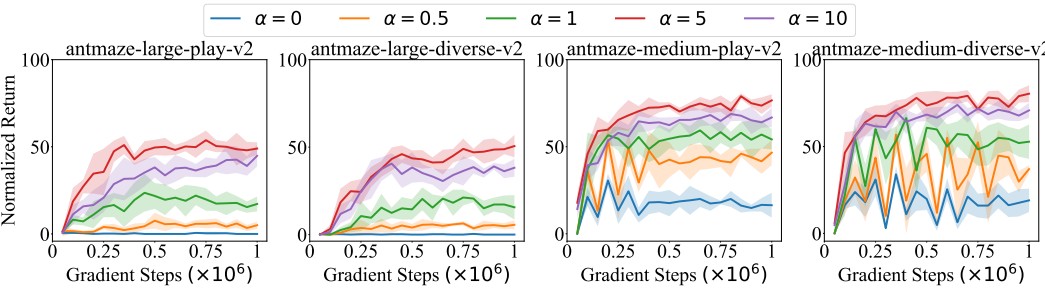

Figure 6: Additional results from the parameter study on the inverse temperature $\alpha$. The curves are averaged over 5 random seeds, with the shaded area representing the standard deviation across seeds.

**Balance Coefficient $\lambda$.** The balance coefficient $\lambda$ controls the balance between vanilla policy improvement and SCAS regularization. If we set $\lambda = 0$, SCAS degenerates into the vanilla off-policy RL algorithm. Here we vary $\lambda$ in $\{0, 0.25, 0.5, 0.75, 1\}$ and present the corresponding learning curves of SCAS in Figure 7. Notably, SCAS is able to converge to good performance over a very wide range of $\lambda$. However, if $\lambda = 0$, the vanilla off-policy RL suffers from extrapolation error and overestimation, demonstrating poor performance. We also observe a very interesting fact that even when $\lambda = 1$ and the signal from RL improvement (max Q) is removed, SCAS still performs well on most tasks. This could be attributed to the fact that value-aware OOD state correction implies some sort of improvement in policy by maximizing the values of policy-induced next states.

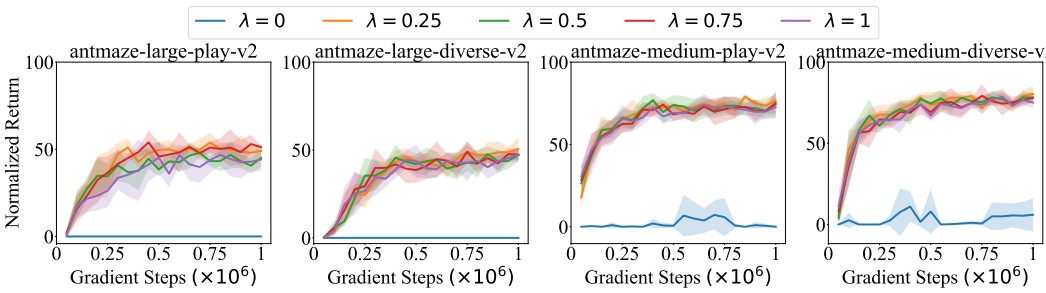

Figure 7: Additional results from the parameter study on the balance coefficient $\lambda$. The curves are averaged over 5 random seeds, with the shaded area representing the standard deviation across seeds.

**Noise Scale $\sigma$.** The noise scale $\sigma$ is the standard deviation of the Gaussian noise added to the original states for formulating the SCAS regularizer. Here we test SCAS with different $\sigma$ and present the corresponding learning curves in Figure 8. We observe a significant performance drop with too large $\sigma$ ($\sigma = 1$) on all the tasks, due to the heavily corrupted learning signal. On the other hand, when $\sigma = 0$ (without noise), the performance is also less satisfying. With $\sigma = 0$, SCAS is still able to prevent the agent at ID states from entering OOD states, maintaining the agent in safe regions, but it cannot correct the agent from OOD states to ID states as reliably as the original SCAS. In general, we find that choosing $\sigma = 0.001$ or $0.01$ leads to the best performance.

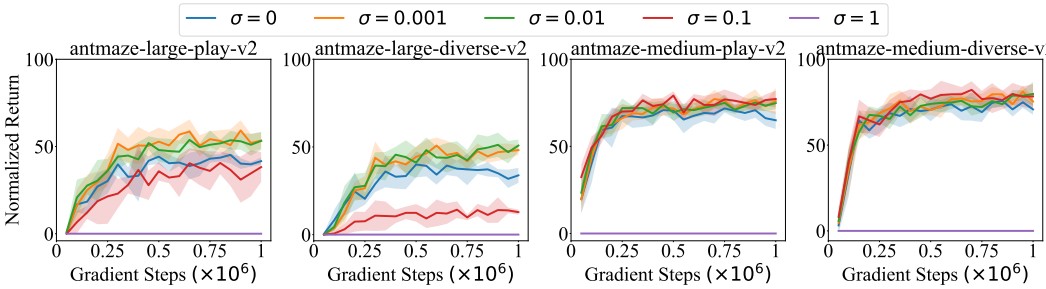

Figure 8: Additional results from the parameter study on the noise scale $\sigma$. The curves are averaged over 5 random seeds, with the shaded area representing the standard deviation across seeds.

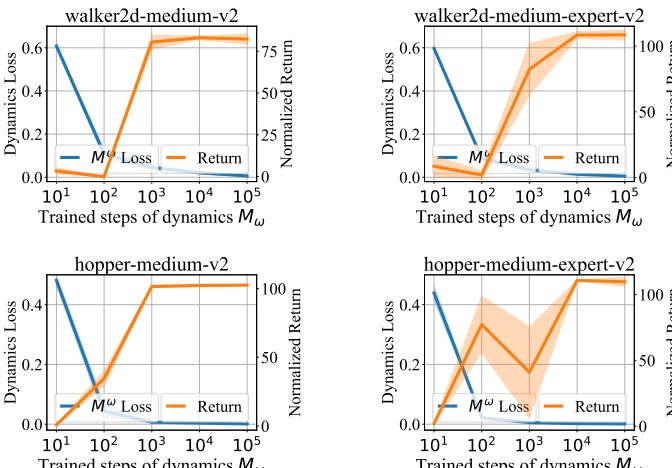

Figure 9: Performance of SCAS under different dynamics model checkpoints, which are obtained at different steps in the dynamics model training process. The figure plots the training loss of the dynamics model $M_\omega$ and the corresponding normalized return of SCAS over 5 random seeds.

### E.7 Sensitivity Analysis on Dynamics Model Errors

To empirically investigate SCAS under different dynamics model errors, we run SCAS using different checkpoints of the trained dynamics model, which are obtained at different steps in the dynamics model training process. The model error is controlled by the number of trained steps. The results are shown in Figure 9. The figure plots the training loss of the dynamics model $M_\omega$ and the corresponding normalized return of SCAS over 5 random seeds. We observe that the performance of SCAS increases with the number of trained steps of the dynamics model (i.e. the accuracy of the model) and stabilizes at a high level.

### E.8 Learning Curves of SCAS

Learning curves on Gym locomotion tasks and Antmaze tasks are presented in Figure 10 and Figure 11 respectively. The curves are averaged over 5 random seeds, with the shaded area representing the standard deviation across seeds.

## F   Broader Impact

Offline RL holds promise for facilitating practical RL applications in domains like robotics, healthcare, and education, where data collection is often costly or risky. However, it is important to recognize its potential negative societal impacts. One concern is that biases in offline data may transfer to the learned policy. In addition, offline RL may affect employment by automating tasks traditionally

performed by humans, like factory automation or autonomous driving. Addressing these challenges will contribute to the responsible development and deployment of offline RL algorithms.

From an academic standpoint, this research systematically analyze the OOD state issue in offline RL and propose SCAS, a simple yet effective approach that unifies OOD state correction and OOD action suppression. This work potentially offers researchers a new perspective on analyzing the OOD state issue and enhancing test-time robustness in offline RL. Besides, SCAS also holds the promise to be extended to safe RL [1, 17, 13], meta RL [8, 65, 66, 64, 4], and multi-agent RL [35, 52, 55, 51, 16].

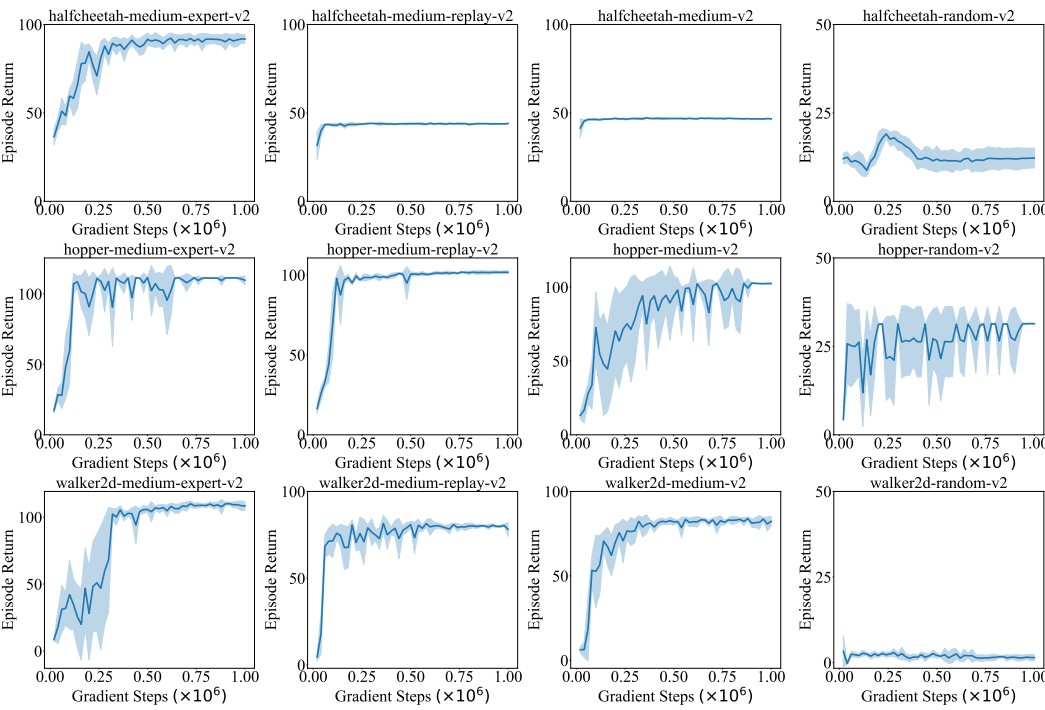

Figure 10: Learning curves of SCAS on Gym locomotion tasks. The curves are averaged over 5 random seeds, with the shaded area representing the standard deviation across seeds.

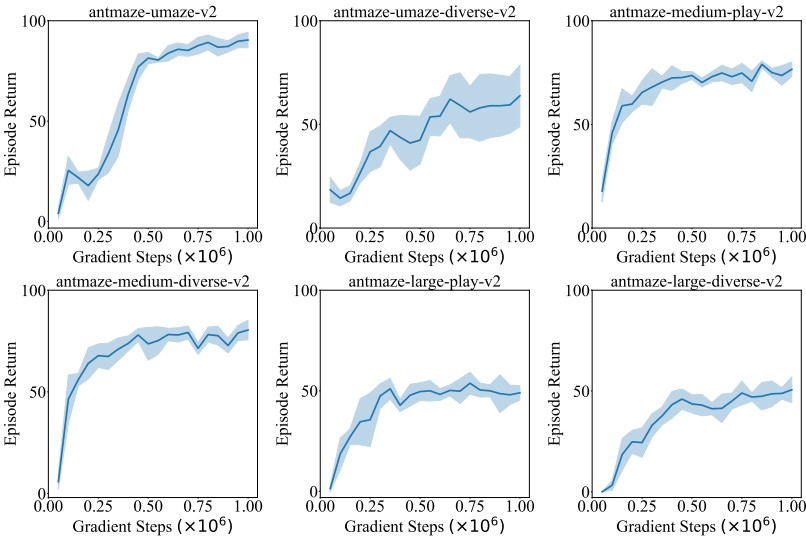

Figure 11: Learning curves of SCAS on AntMaze tasks. The curves are averaged over 5 random seeds, with the shaded area representing the standard deviation across seeds.

