# OpenReview forum: "Offline Reinforcement Learning with OOD State Correction and OOD Action Suppression"
_NeurIPS.cc/2024/Conference — NeurIPS 2024 poster_

### Official Review · Reviewer_fu1b · 2024-06-29

**Soundness:** 4
**Presentation:** 2
**Contribution:** 3
**Rating:** 6
**Confidence:** 3

**Summary:**

The issue of OOD actions is a well-known concern in offline RL research. The issue of OOD states, however, is relatively unexplored. In this paper, the authors shed light on the significance of OOD state correction and propose a SCAS algorithm to guide the agent back into high-value ID states when encountering an OOD state. To achieve this goal, SCAS first defines a conditional state transition probability $N^*(s'\mid s)$ skewed towards high-valued states and regularizes the policy by aligning the dynamics induced by the policy on a perturbed state $\hat{s}\sim \mathcal{N}(s, \sigma^2I)$ with the value aware state transition probability $N^*(\cdot \mid s)$. Also, theoretical analysis shows that SCAS implicitly guarantees the correction of OOD actions, thereby removing the need for additional regularizers.

**Strengths:**

The authors highlighted the significance of OOD state correction in offline RL, a problem other researchers often neglect. Also, multiple experiments were conducted to show the competence of SCAS and its robustness towards the choice of hyperparameters.

**Weaknesses:**

1. The authors did not cite works from the DICE literature(e.g., Nachum et al., 2019; Lee et al., 2021; Mao et al., 2024). Although they do not explicitly correct OOD states, I believe they are deeply related to SCAS because they aim to align the policy's state distribution $d^\pi$ with the dataset's state distribution, thus preventing the policy from visiting OOD states.

2. The choice of $\exp(\alpha V(s))$ as the empirical normalizer seems extremely arbitrary. Providing some rationale behind this choice is recommended.

3. All of the experimental analysis was performed on MuJoCo environments, which have deterministic transition dynamics. Since SCAS utilizes a deterministic environmental model, it may not perform well in environments with stochastic transition dynamics.

4. NeoRL is a relatively uncommon benchmark compared to D4RL. A brief explanation of NeoRL would be helpful for the readers.

5. Some experimental details are missing from the paper.

    * Do Figures 1a to 1d share the same embedding function? How was this embedding function obtained? Was it by running t-SNE on the set of 200,000 samples(50,000 samples each from the dataset, CQL, TD3+BC, and SCAS)?
    * The points in Figure 1d seem to be the union of the points in Figures 1a, 1b, and 1c. Am I correct?
    * Why does Figure 2 omit Q values of off-policy RL, SDC, and OSR for higher numbers of optimization steps?
    * Section 6.3 states that varying steps of Gaussian noise were added to the actions during test time. Figure 3 indicates that the authors added noise up to 40 steps. However, trajectories are usually longer than 40 steps. How were those 40 steps selected? Did the authors add noise to the first 40 interactions with the environment?

6. Minor comment: The meaning of the term *$\mathcal{D}(s)$ weights* on Line 162 is unclear.


### References

Lee, Jongmin, et al. "Optidice: Offline policy optimization via stationary distribution correction estimation." International Conference on Machine Learning. PMLR, 2021.

Mao, Liyuan, et al. "Odice: Revealing the mystery of distribution correction estimation via orthogonal-gradient update." arXiv preprint arXiv:2402.00348 (2024).

Nachum, Ofir, et al. "Algaedice: Policy gradient from arbitrary experience." arXiv preprint arXiv:1912.02074 (2019).

**Questions:**

Please refer to the **Weaknesses** section. As a side note, have you tried using the Q and V functions learned by IQL (or any other value-based offline RL algorithm) instead of learning them together with the policy? From Figure 9, the training process seems a bit unstable, so using a stationary training objective might help.

**Limitations:**

The authors have adequately addressed their work's limitations and potential negative societal impact.

---

> ### Author Rebuttal · Authors · 2024-08-07
>
> We appreciate the time and effort you are dedicated to providing feedback on our paper and are grateful for the meaningful comments.
>
> **Q1: The authors did not cite works from the DICE literature.**
>
> Thanks for the suggestion. We will cite and add discussions on DICE literature as follows.
>
> Although the DICE series of works [1,2,3] share similar motivations with SCAS, there are significant differences between the two. Firstly, DICE is based on a linear programming framework of RL, while SCAS is based on a dynamic programming framework. Therefore, the theoretical foundations and learning paradigms of the two are inherently different. Secondly, SCAS only corrects encountered OOD states, whereas DICE algorithms require the policy-induced occupancy distribution $d^\pi$ to align with the dataset distribution $d^\mathcal D$. Therefore, DICE's constraints are stricter, potentially making it more susceptible to the average quality of the dataset. Lastly, theoretical and empirical evidence indicate that DICE algorithms have a problem of gradient cancellation [1], which imposes certain limitations on their practical effectiveness.
>
> **Q2: The choice of $\exp(\alpha V(s))$ as the empirical normalizer seems extremely arbitrary. Providing some rationale behind this choice is recommended.**
>
> Thanks for the suggestion. Firstly, choosing $\exp(\alpha V(s))$ is intended to obtain something similar to the advantage function. With this normalizer, the weight of our regularizer is $\exp(\alpha (V(s') - V(s)))$, which is comparable to the weight $\exp(\alpha A(s,a))$ in Advantage Weighted Regression (AWR) [4]. Here, $V(s') - V(s)$ represents the relative advantage of the next state $s'$ compared to the current state $s$, while $A(s,a)$ reflects the relative advantage of taking action $a$ in $s$ compared to following the current policy. Comparison of the objectives of SCAS and AWR:
>
> SCAS: $\exp(\alpha (V(s’)-V(s))) \log(M(s’|\hat s,\pi(\hat s)))$;
>
> AWR: $\exp(\alpha A(s,a)) \log\pi(a|s)$.
>
> Secondly, as discussed in the paper (lines 159-166), introducing any normalizer that depends only on $s$ (independent of $s'$) does not affect the development and analysis of our method; it is merely for computational stability. In AWR-based methods, there also exists a normalizer $Z(s)$ and they usually disregard it. The rationale behind this is similar.
>
> **Q3: Since SCAS utilizes a deterministic environmental model, it may not perform well in environments with stochastic transition dynamics.**
>
> Thanks for the comment. Although the implementation of SCAS uses a deterministic dynamics model, the SCAS framework is compatible with stochastic dynamics models. The reasons to employ deterministic models are (1) our considered environments retain in MuJoCo environments, which are mostly deterministic in state transitions; and (2) deterministic models are easy to train and implement in practice. We also admit that deterministic dynamics modeling might encounter more errors when the transition dynamics are multimodal. As mentioned in the limitation section, we expect to find a better dynamics model to further enhance our method in such scenarios.
>
> **Q4: A brief explanation of NeoRL would be helpful for the readers.**
>
> Thanks for the kind suggestion. We will include the introduction as follows.
>
> NeoRL is a benchmark designed to simulate real-world scenarios by collecting datasets using a more conservative policy, aligning closely with realistic data collection scenarios. The narrow and limited data makes it challenging for offline RL algorithms.
>
> **Q5: Some experimental details are missing from the paper.**
>
> Thanks for the comments and we apologize for any confusion resulting from the missing details.
>
> > Do Figures 1a to 1d share the same embedding function? How was this embedding function obtained? Was it by running t-SNE on the set of 200,000 samples(50,000 samples each from the dataset, CQL, TD3+BC, and SCAS)?
>
> Yes. Figures 1a to 1d share the same embedding function obtained by running t-SNE on the set of all 200,000 samples (50,000 samples each from the dataset, CQL, TD3+BC, and SCAS). This ensures a clear visual comparison.
>
> > The points in Figure 1d seem to be the union of the points in Figures 1a, 1b, and 1c.
>
> Yes. Figure 1d contains all the 200,000 samples, which is the union of the points in Figures 1a, 1b, and 1c.
>
> > Why does Figure 2 omit Q values of off-policy RL, SDC, and OSR for higher numbers of optimization steps?
>
> Sorry for the confusion. Because the Q values of Off-policy RL, SDC w/o CQL, and OSR w/o CQL diverge in the early learning stage, plotting their Q values at later optimization steps would result in an excessive range on the vertical axis. For clearer presentation, we have replotted this figure in **Figure 2** of the PDF (attached to the global response), which adds more data points to the early learning stage.
>
> > How were those 40 steps selected? Did the authors add noise to the first 40 interactions with the environment?
>
> We choose a maximum of 40 steps for perturbations because we empirically find that, under the noise magnitude of 0.5, adding more than 50 steps of noise often causes the agent to directly enter a terminal state during the perturbation, making it difficult to assess the agent's ability to correct itself from OOD states after the perturbation. Given that the episode length of our tasks is 1000 steps, we start applying perturbations at step 500 during test time. This choice is because, at the beginning, when the agent is still in its initial state and not yet in motion, applying perturbations is essentially like changing the initial state, which is not very meaningful. In contrast, applying perturbations at step 500 helps in evaluating the agent's robustness to perturbations while it is in motion.
>
>  We will add these explanations to the latter revision.
>
> **Due to the page limit, please refer to the next block. Thanks!**

---

> > ### Comment · Reviewer_fu1b · 2024-08-09
> >
> > Thank you for your response. All of my concerns were resolved, so I have updated my score.

---

> > > ### Author Response · Authors · 2024-08-09
> > >
> > > Thank you for your positive feedback! We truly appreciate the time and effort you devoted to reviewing our manuscript. Your comments and suggestions are very helpful.

---

> ### Author Response · Authors · 2024-08-07
> **Additional rebuttal contents**
>
> ------Thank you for continuing to read!------
>
> **Q7: Minor comment: The meaning of the term $\mathcal D(s)$ weights on Line 162 is unclear.**
>
> Thanks for pointing that out. Here, $D(s)$ indicates the offline dataset's empirical state distribution.
>
> **Q8: Have you tried using the Q and V functions learned by IQL (or any other value-based offline RL algorithm) instead of learning them together with the policy?**
>
> Yes, when implementing the algorithm, we experimented with using IQL to train the value function. The results of this variant, IQL+SCAS, are shown in **Table 2** of the PDF. However, we found that IQL+SCAS does not perform better than the original SCAS, which uses vanilla policy evaluation. The performance of IQL+SCAS is comparable on most tasks and slightly worse on some tasks. We have two hypotheses: (1) The policy and value training objectives in IQL+SCAS are not well aligned, which is a known issue with IQL identified in [5]; and (2) SCAS already has the effect of suppressing OOD actions and using IQL to train the value function might introduce additional conservatism.
>
> **Reference**
>
> [1] Lee, Jongmin, et al. "Optidice: Offline policy optimization via stationary distribution correction estimation." ICML 2021.
>
> [2] Mao, Liyuan, et al. "Odice: Revealing the mystery of distribution correction estimation via orthogonal-gradient update." ICLR 2024.
>
> [3] Nachum, Ofir, et al. "Algaedice: Policy gradient from arbitrary experience." arXiv preprint arXiv:1912.02074 (2019).
>
> [4] Peng, Xue Bin, et al. "Advantage-weighted regression: Simple and scalable off-policy reinforcement learning." arXiv preprint arXiv:1910.00177 (2019).
>
> [5] Xu, Haoran, et al. "Offline rl with no ood actions: In-sample learning via implicit value regularization." ICLR 2023.

---

### Official Review · Reviewer_VyLC · 2024-07-10

**Soundness:** 3
**Presentation:** 2
**Contribution:** 2
**Rating:** 6
**Confidence:** 4

**Summary:**

This paper presents SCAS (OOD State Correction and Action Supression), a model-based regularization approach that effectively addresses the challenges of out-of-distribution (OOD) states and actions in offline reinforcement learning (RL) algorithms.
The method unfolds in two main stages: (1) training a transition model, and (2) training a policy that incorporates a model-based regularizer.
This regularizer is designed to steer the policy towards high-value in-distribution (ID) successor states and thereby away from OOD successor states.
While the primary focus of SCAS is to resolve OOD state issues, this paper shows that the regularization strategy not only mitigates visiting OOD states but, as a byproduct, also suppresses OOD actions in the training of the value function.
Empirical results demonstrate that the proposed method performs comparably on standard benchmarks, showing its effectiveness in refining offline RL algorithms.

**Strengths:**

**1. Theoretical connection between OOD state correction and OOD action suppression**

A notable strength of this paper is the establishment of a theoretical link between OOD state correction and OOD action suppression. The authors articulate how the mechanism designed to correct OOD states implicitly suppresses OOD actions, by showing an optimal policy of the regularized objective will produce actions inside of the support of behavior policy for every dataset state. This implicit suppression also explains successful empirical results on offline RL benchmarks.

**2. Robustness to Hyperparameter Selection**

As detailed in Section 6.2 and illustrated in Figure 4, the proposed approach demonstrates considerable robustness to variations in hyperparameter settings.
This attribute is particularly valuable in practical applications of offline RL where optimal hyperparameter settings can be elusive or computationally expensive to determine.

**3. Evaluation on standard benchmarks and implementation code is attached for reproduction.**

The paper's evaluation methodology is another major strength. The authors have rigorously tested their approach on standard benchmarks, D4RL and NeoRL, providing a comprehensive assessment of its performance relative to existing methods. Moreover, the inclusion of implementation code enhances the paper’s contribution by facilitating reproducibility and further experimentation.

**Weaknesses:**

**1. Unclear Motivation for Value-Aware OOD State Correction**

This paper proposes shifting the OOD state distribution not to a standard ID state distribution, but to a high-value ID state distribution instead. This choice raises questions about the specific objectives of value-aware state correction. Is it strategically targeting high-value data points, or is it intended to mitigate the effects of distribution shifts? It seems that the manuscript primarily focuses on the former, in that case, it would benefit from a comparison with existing works like DW[1], which also focus on high-value data points for behavior regularization. If the goal is indeed to mitigate the effects of distribution shifts, then it is crucial for the manuscript to better articulate and emphasize how value-aware OOD state correction contributes to the robustness against OOD state visitation. Moreover, a clearer explanation of how this approach aligns with offline RL principles like pessimism and robustness to OOD states would greatly strengthen the manuscript's validity and impact.

**2. Lack of Comparisons with Comparable Offline RL Approaches**

While the paper tackles a fundamental problem in offline RL, the evaluations presented are somewhat constrained by a lack of comparison with key recent advancements in the field. Notable works such as DW[1], EDAC[2], RORL[3], and SQL/EQL[4], which explore similar offline RL settings, are not considered in the main experiments. This gap is particularly significant as DW[1] employs a value-aware strategy that closely mirrors the approach of this paper, focusing on constraining policies to high-value data points. To convincingly justify the need for correcting OOD states—or for employing a value-aware correction approach—a comparative analysis with these influential studies is essential. This would not only place the findings in a broader context but also potentially highlight the contributions and advantages of the proposed method.

[1] Hong et al., “Beyond Uniform Sampling: Offline Reinforcement Learning with Imbalanced Datasets.”, NeurIPS 2023.

[2] An et al., "Uncertainty-Based Offline Reinforcement Learning with Diversified Q-Ensemble.", NeurIPS 2021.

[3] Yang et al., “RORL: Robust Offline Reinforcement Learning via Conservative Smoothing.”, NeurIPS 2022.

[4] Xu et al., “Offline RL with No OOD Actions: In-Sample Learning via Implicit Value Regularization.”, ICLR 2023.

**Questions:**

Q1. How sensitive is the performance to the accuracy of the model?

Q2. Given that a main contribution of this work is a model-based regularizer, it appears that SCAS could be compatible with various offline RL methods. Is it feasible to apply the regularizer across different offline RL objectives, such as CQL, IQL, and others?

Q3. In this study, the trained model is exclusively utilized as a regularizer during policy training. Could this approach be extended to also incorporate the model during testing time?

**Limitations:**

The authors included their limitations in Section 7 and limitations are addressed appropriately.

---

> ### Author Rebuttal · Authors · 2024-08-07
>
> We appreciate the time and effort you are dedicated to providing feedback on our paper and are grateful for the meaningful comments. We have conducted extensive experiments to address your questions and concerns.
>
> **Q1: Unclear Motivation for Value-Aware OOD State Correction.**
>
> > Is it strategically targeting high-value data points, or is it intended to mitigate the effects of distribution shifts?
>
> In this work, (i) the mitigation of state distribution shift, (ii) the mitigation of action distribution shift, and (iii) the tendency towards high-value data points are seamlessly integrated into a single framework (regularizer), whereas previous OOD state correction works only consider (i).
>
> Value-aware OOD state correction is a concept introduced in the manuscript. Since previous work has already considered (i), its motivation is to additionally achieve (iii). We demonstrate that within the SCAS framework, value-aware OOD state correction can be easily implemented, and it also implicitly achieves (ii). It is worth noting that SCAS with $\alpha=0$ (vanilla OOD state correction in SCAS's framework, which differs from previous works) still has the effects of (i) and (ii). Compared to it, the motivation behind value-aware OOD state correction is primarily to achieve a tendency towards high-value data points, rather than to further improve the mitigation of distribution shifts. We do not claim that SCAS with $\alpha>0$ is more effective in mitigating distribution shifts compared to SCAS with $\alpha=0$.
>
> In the following, we detail why SCAS not only mitigates state and action distribution shifts but also targets high-value data points.
>
> Mitigating distribution shifts: From Eq. (5) and Eq. (6), $N^*(\cdot|s)$ lies within the support of the dataset. Therefore, when aligning the dynamics induced by $\pi$ on the perturbed state $\hat{s}$ with $N^*(\cdot|s)$ in Eq. (8), the result is that $\pi$-induced $s'$ also lies within the dataset support. On the other hand, theory (Propositions 1 and 2) and experiments (Section 6) indicate that, in ID states, the actions outputted by $\pi$ also lie in the dataset support. Therefore, the SCAS framework naturally mitigates both state and action distribution shifts, which is a key feature of our design.
>
> Targeting high-value data points: From Eq. (5) and Eq. (6), $N^*(s)$ is skewed towards high-value ID states. As a result, $\pi$-induced $s'$ is not only corrected back to the dataset support but also guided towards high-value states. Therefore, SCAS is also strategically targeting high-value ID states.
>
> > it would benefit from a comparison with existing works like DW[1], which also focus on high-value data points for behavior regularization.
>
> Thanks for the kind suggestion. We've taken your advice to include DW [1] in comparison. Results are presented in **Table 1** of the PDF (attached to the global response). DW reweights ID data points by their values for behavior regularization and does not account for OOD states during the test phase. In contrast, our approach considers an OOD state correction scenario, resulting in enhanced robustness during the test phase and better performance in both MuJoCo locomotion and AntMaze domains. We will discuss DW and include the comparison results in the latter revision.
>
> > Moreover, a clearer explanation of how this approach aligns with offline RL principles like pessimism and robustness to OOD states would greatly strengthen the manuscript's validity and impact.
>
> Thanks for the suggestion. In a specific sense, SCAS, which unifies OOD state correction and OOD action suppression, also integrates pessimism and state robustness. (1) Regarding pessimism: The OOD action suppression effect of SCAS aligns with the pessimism commonly discussed in offline RL work (being pessimistic about OOD actions). Unlike traditional policy constraint methods, our approach does not require the training policy to align with the behavior policy; it only requires the successor states to be within the dataset support, which is a more relaxed constraint. (2) Regarding state robustness: The OOD state correction effect of SCAS is aimed at improving the agent's robustness to OOD states during the test phase. Compared with previous works, SCAS unifies OOD state correction and OOD action suppression and additionally achieves value-aware OOD state correction. Some offline RL literature on state robustness differs from our approach; they typically consider noisy observations [2], such as sensor errors. In contrast, SCAS addresses state robustness concerning actual OOD states encountered during test time, rather than noisy observations.
>
> **Q2: Lack of Comparisons with Comparable Offline RL Approaches.**
>
> Thanks for the suggestion. Note that the original SCAS requires only one single hyperparameter configuration in implementations. For a fair comparison with DW[1]/EDAC[3]/RORL[2]/SQL[4]/EQL[4], we roughly select $\lambda$ from {0.025, 0.25} for each dataset, referring to this variant as SCAS-ht. The results of SCAS-ht and these methods are reported in **Table 1** of the PDF. Among the ensemble-free methods, SCAS-ht achieves the highest performance in both mujoco locomotion and antmaze domains. Compared with ensemble-based methods, SCAS-ht also performs better on antmaze tasks. We will include the comparison results in the latter revision.
>
> **Due to the page limit, please refer to the next block. Thanks!**

---

> ### Author Response · Authors · 2024-08-07
> **Additional rebuttal contents**
>
> ------Thank you for continuing to read!------
>
> **Q3: How sensitive is the performance to the accuracy of the model?**
>
> Thanks for your question. To empirically investigate SCAS under different dynamics model errors, we run SCAS using different checkpoints of the trained dynamics model, which are obtained at different steps in the dynamics model training process. The model error is controlled by the number of trained steps. The results are shown in **Figure 1** of the PDF. The figure plots the training loss of the dynamics model $M_\omega$ and the corresponding normalized return of SCAS over 5 random seeds. We observe that the performance of SCAS increases with the number of trained steps of the dynamics model (i.e. the accuracy of the model) and stabilizes at a high level.
>
> **Q4: Is it feasible to apply the regularizer across different offline RL objectives, such as CQL, IQL, and others?**
>
> Yes, the SCAS regularizer is compatible with various offline RL methods. We have conducted experiments to combine SCAS with CQL, IQL, and TD3BC. Comparisons between these combined algorithms and the original ones are shown in **Table 2** of the PDF. We find that applying the SCAS regularizer leads to improved performance for all these popular algorithms, which could be attributed to the OOD state correction effects of SCAS. However, we also find that these combined methods do not achieve better performance than the original SCAS (comparable on most tasks and worse on some tasks). We hypothesize that this is because SCAS already has the effect of OOD action suppression, and when combined with offline RL objectives that also aim for OOD action suppression, it may become overly conservative. As a result, the combined algorithms may perform worse than the original SCAS on some sub-optimal datasets.
>
> **Q5: Could this approach be extended to also incorporate the model during testing time?**
>
> This is a good perspective! Indeed, SCAS utilizes the trained model as a regularizer during policy training. Incorporating the model during testing time for OOD state correction can be an alternative approach (or an enhancement to SCAS) to achieving test-time OOD state correction. These two directions appear to be completely different in algorithmic designs, and it would be an interesting direction for future work. Thanks for the insightful comment.
>
> **Reference**
>
> [1] Hong, Zhang-Wei, et al. "Beyond uniform sampling: Offline reinforcement learning with imbalanced datasets." NeurIPS 2023.
>
> [2] Yang, Rui, et al. "Rorl: Robust offline reinforcement learning via conservative smoothing." NeurIPS 2022.
>
> [3] An, Gaon, et al. "Uncertainty-based offline reinforcement learning with diversified q-ensemble." NeurIPS 2021.
>
> [4] Xu, Haoran, et al. "Offline rl with no ood actions: In-sample learning via implicit value regularization." ICLR 2023.

---

> > ### Comment · Reviewer_VyLC · 2024-08-11
> > **Response to Author Rebuttal**
> >
> > Thank you for your insightful comments and for conducting the extensive set of additional experiments!
> >
> > Since my primary concern regarding the lack of experimental comparisons has been fully addressed, I have raised my rating.
> >
> > This work now presents (1) promising empirical results and (2) a generic model-based regularizer that can be easily integrated into existing offline RL algorithms. Hence, I believe that these contributions are substantial for the advancement of the offline RL community.

---

> > > ### Author Response · Authors · 2024-08-11
> > >
> > > Thank you for your positive feedback! We sincerely appreciate the time and effort you have dedicated to reviewing our manuscript. Your comments and suggestions are highly valuable to us, and we will carefully incorporate them into the latter revision.

---

### Official Review · Reviewer_VgaK · 2024-07-11

**Soundness:** 2
**Presentation:** 3
**Contribution:** 3
**Rating:** 6
**Confidence:** 3

**Summary:**

This paper focuses on OOD state issue, an important but overlooked issue in offline RL. This paper proposes aligning the OOD state towards In-Distribution (ID) states with high value, named as value-aware OOD state correction. Additionally, the paper discovers that the overestimation of OOD actions can also be mitigated during the implementation of OOD state correction. The experimental results demonstrate that the proposed algorithm is superior in performance and robustness.

**Strengths:**

1. The paper presents an intuitive solution to the OOD state iusse, while also mitigating the overestimation of OOD actions.

2. The algorithm is highly robust, requiring only a single set of hyperparameters across different environments. This robustness is particularly important in the context of offline reinforcement learning, as online interaction for tuning parameters is costly or even dangerous.

**Weaknesses:**

1. The paper proposes that during the process of OOD state correction, the overestimation of OOD actions will also be addressed. I have some doubts about this point. The constraint $R_2$ seems to only regularize the policy to output actions at OOD states that result in the next state being within the offline dataset. However, there is no regularization on the policy's output actions at ID states. Therefore, I am confused about how the paper solves the traditional issue with OOD actions. If I have misunderstood anywhere, please point it out.

**Questions:**

1. Figure 2 suffers from an excessive range on the vertical axis, causing the two curves of SCAS to almost overlap. A suggestion would be to take the log of the Q values on the vertical axis; this should make the two curves more distinguishable.

**Limitations:**

Please see weakness.

---

> ### Author Rebuttal · Authors · 2024-08-07
>
> We appreciate the time and effort you are dedicated to providing feedback on our paper and are grateful for the meaningful comments.
>
> **Q1:  There is no regularization on the policy's output actions at ID states. How the paper solves the traditional issue with OOD actions?**
>
> We apologize for the confusion. Actually, in our method, there is regularization on the policy's output actions at ID states. In our regularizer $R_2$, the perturbed states $\hat{s}$ are sampled from $\mathcal{N}(s,\sigma^2)$, and a large portion of $\hat{s}$ will fall near the original ID state $s$ or even be approximately equal to $s$. Therefore, the policy's output actions at ID states are also regularized in $R_2$. For this part of the regularization, its role is equivalent to the ID state regularizer analyzed in Section 4, which has been theoretically shown to have the effect of suppressing OOD actions. Moreover, the experimental results in Section 5 also demonstrate that our OOD state correction regularizer addresses the traditional issue with OOD actions.
>
> **Q2: Figure 2 suffers from an excessive range on the vertical axis. A suggestion would be to take the log of the Q values on the vertical axis.**
>
> Thanks for pointing that out and for the advice. Actually, we already use a log scale for the vertical axis in Figure 2. To further address this excessive range issue, we have replotted this figure in **Figure 2** of the PDF (attached to the global response), which adds more data points to the early learning stage and uses a smaller vertical axis range.

---

> > ### Comment · Reviewer_VgaK · 2024-08-11
> >
> > Thank you for addressing my confusion. Incorporating the description for $R2$ and the refined figures into the future version would further improve the readability of the paper.
> >
> > Taking into account the supplementary experiments that validate the generalizability of the approach, I have raised my score to support the acceptance of the paper.

---

> > > ### Author Response · Authors · 2024-08-11
> > >
> > > Thank you for your positive feedback! We sincerely appreciate the time and effort you have dedicated to reviewing our manuscript. Your comments and suggestions are highly valuable to us, and we will carefully incorporate them into the latter revision.

---

### Official Review · Reviewer_j7HR · 2024-07-13

**Soundness:** 3
**Presentation:** 3
**Contribution:** 3
**Rating:** 6
**Confidence:** 4

**Summary:**

The paper proposes a regularization term in offline RL, which can simultaneously address OOD state correction and OOD action suppression without pretraining a state transition model $N(s'|s)$. It shows good performance in D4RL benchmarks.

**Strengths:**

I found it an interesting topic to consider the OOD state issue. Most existing offline RL methods focus on addressing the OOD action issue, while this paper highlights that OOD states may also cause training collapse. Another good property of the method is that it only needs one single hyperparameter configuration to achieve good empirical performance. The theory is also solid.

**Weaknesses:**

The method utilizes a deterministic policy, which is often regarded as lacking expressiveness. Thus, the performance is not as good as diffusion-based policy methods.

**Questions:**

1. For Definition 1, if $d^\pi_{M_\tau}$ is a continuous probability density function of $s$, and if $d_D(s)$ only has support on $s$ which appears in the dataset, does it mean that any state which doesn't show up in the dataset is OOD? If that's the case, the probability of an OOD state is 1 since the number of ID states must be finite in the dataset?

2. Instead of pretraining $N(s'|s)$, the paper chooses to add Gaussian noise to $s$, with the noise $\sigma=0.003$. Is this hyperparameter important? If $\sigma$ increases, can we have a more robust algorithm since it makes more OOD states into consideration?

3. The method uses a deterministic policy. What if we choose a Gaussian policy?

4. In Figure 2, why do Off-policy, SDC, and OSR only have one dot? Shouldn't it also be a line?

5. How are the results in Table 1 recorded? Final round results, last 5 epoch average, or another method?

6. Where do the baseline method results come from, their original paper? I ask because I found SDC doesn't have results for antmaze tasks. Did the authors implement it by themselves?

7. The authors use antmaze-v2 for their method evaluation, while some baseline methods, for example, IQL and OSR, use antmaze-v0. In my experience, the v2 dataset can always provide higher rewards than v0. Can the authors explain the mismatch? Or can the authors show results on v0 as well?

**Limitations:**

I believe the theory of the paper is solid, while the experimental performance is my concern. I am happy to increase my score if the antmaze version mismatch issue is resolved.

---

> ### Author Rebuttal · Authors · 2024-08-07
>
> We appreciate the time and effort you are dedicated to providing feedback on our paper and are grateful for the meaningful comments.
>
> **Q1: About antmaze version.**
>
> Special thanks for your careful review. Actually, all the antmaze results presented in the paper are obtained from the **antmaze-v2** datasets. We noted that some baseline methods report results from antmaze-v0 (or not include antmaze experiments) in their original papers. For the results reported in Table 1 of the paper, we reran OSR, SDC, OneStep RL, and BC on antmaze-v2 datasets and took other baselines' antmaze-v2 results from the SPOT paper[2].
>
> To answer your question more comprehensively, we have also run SCAS on the antmaze-v0 datasets. Since IQL has demonstrated superior performance compared to other baselines on the v2 datasets, we present a comparison of SCAS and IQL on antmaze-v0 in the following table. The results show that SCAS also outperforms IQL on antmaze-v0 tasks.
>
> Table: Comparison of SCAS and IQL on antmaze-v0 over 5 random seeds.
>
> | Dataset | IQL | SCAS |
> | --- | --- | --- |
> | antmaze-umaze-v0 | 87.5 | **90.4$\pm$3.6** |
> | antmaze-umaze-div-v0 | 62.2 | **66.4$\pm$14.3** |
> | antmaze-med-play-v0 | 71.2 | **76.4$\pm$4.0** |
> | antmaze-med-div-v0 | 70.0 | **76.0$\pm$3.2** |
> | antmaze-large-play-v0 | 39.6 | **45.6$\pm$4.8** |
> | antmaze-large-div-v0 | **47.5** | **47.2$\pm$7.2** |
> | antmaze-v0 total | 378.0 | **402.0** |
>
> **Q2: The method utilizes a deterministic policy, which is often regarded as lacking expressiveness. Thus, the performance is not as good as diffusion-based policy methods.**
>
> Thanks for the meaningful comments. We address the concerns from three aspects.
>
> (1) About performance. One important reason why SCAS's performance might appear less impressive compared to diffusion-based policy methods is that SCAS uses only a single hyperparameter configuration. With slight parameter tuning, SCAS's performance can also be significantly improved. We roughly select $\lambda$ from {0.025, 0.25} for each dataset, referring to this variant as SCAS-ht. Comparisons of SCAS-ht with Diffusion QL [1] and additional recent algorithms are presented in **Table 1** of the PDF (attached to the global response). Among the ensemble-free methods, SCAS-ht achieves the highest performance in both mujoco locomotion and antmaze domains. Compared with ensemble-based methods, SCAS-ht also performs better on antmaze tasks.
>
> (2) About deterministic policy. Diffusion-based policies are highly expressive, allowing them to better represent multi-modal distributions such as behavior policies. However, some work also hypothesizes that deterministic policy may be more suitable than stochastic policy for representing the optimal policy in offline RL [2]. To answer this question in the context of SCAS, we have also included experiments with Gaussian policies. The results and analysis are detailed in our response to Q5. In addition, compared with diffusion-based policies, the deterministic one is computationally efficient as no sampling procedure about latent variables is required. This makes SCAS more suitable for time-sensitive decision-making scenarios.
>
> (3) About compatibility. The SCAS framework does not conflict with diffusion-based policies, and exploring this integration would be an interesting direction for future work.
>
> **Q3: For Definition 1, if $d^\pi_{M_\mathcal T}$ is a continuous probability density function of $s$, and if $d_\mathcal{D}(s)$ only has support on $s$ which appears in the dataset, does it mean that any state which doesn't show up in the dataset is OOD?**
>
> In this paper, we refer to in-dataset states as ID and states outside the dataset as OOD for convenience. According to this definition of OOD states, the probability of an OOD state in a continuous space is indeed 1. For a more general mathematical definition, $d_\mathcal{D}(s)$ in Definition 1 should be replaced with $d^\beta_\mathcal{M}$, where $\beta$ and $\mathcal{M}$ are the behavior policy and environment used to collect the dataset. Under this definition of OOD states, the probability of an OOD state in a continuous space is no longer 1. Essentially, $d_\mathcal{D}(s)$ is the empirical distribution obtained by sampling from $d^\beta_\mathcal{M}$, and the development and analyses of the method in this paper remain unaffected.
>
> **Q4: Is this hyperparameter ($\sigma$) important? If $\sigma$ increases, can we have a more robust algorithm since it makes more OOD states into consideration?**
>
> Thanks for this question. The ablation results on $\sigma$ are presented in Fig. 8 in the Appendix. The results show that despite considering more OOD states, too large $\sigma$ leads to a significant performance drop, due to the heavily corrupted learning signal. On the other hand, when $\sigma=0$ (without noise), the performance is also less satisfying. With $\sigma=0$, SCAS can still suppress OOD actions, but it cannot correct the agent from OOD states to ID states as reliably as the original SCAS. In general, we find that choosing $\sigma$ within the range {0.001, 0.01} leads to the best performance.
>
> **Q5: The method uses a deterministic policy. What if we choose a Gaussian policy?**
>
> Thanks for the question. We implement a version of SCAS with Gaussian policy and report its results in **Table 2** of the PDF. Overall, the performance of SCAS-Gaussian is comparable to the original SCAS on most tasks but is slightly worse on some tasks. We hypothesize that there may be two reasons for this: (1) Stochastic policy optimizes a lower bound of Eq. (10) while deterministic policy ensures the equality case. (2) Deterministic policy may, empirically, be more suitable than Gaussian policy in offline RL [2].
>
> **Due to the page limit, please refer to the next block. Thanks!**

---

> ### Author Response · Authors · 2024-08-07
> **Additional rebuttal contents**
>
> ------Thank you for continuing to read!------
>
> **Q6: In Figure 2, why do Off-policy, SDC, and OSR only have one dot? Shouldn't it also be a line?**
>
> We apologize for the confusion. Because the Q values of these methods diverge in the early learning stage, showing the complete lines would result in an excessive range on the vertical axis. For clearer presentation, we have replotted this figure in **Figure 2** of the PDF, which adds more data points and shows the lines in the early learning stage.
>
> **Q7: How are the results in Table 1 recorded? Final round results, last 5 epoch average, or another method?**
>
> Our evaluation criteria follow those used in most previous works. We evaluated the last trained policy, with each experiment repeated over 5 random seeds. For the Gym locomotion tasks, we average returns over 10 evaluation trajectories, while for the Antmaze tasks, we average over 100 evaluation trajectories.
>
> **Q8: Where do the baseline method results come from?**
>
> The baseline method results are obtained as follows. We re-run OSR on all datasets using their official codebase and tune the hyperparameters for each dataset as specified in their paper. We implement SDC and re-run it on all datasets. We tune the SDC-related hyperparameters as specified in their paper, and sweep the CQL-related hyperparameters in {1,2,5,10,20} for each dataset. We re-run OneStep RL on all datasets using its official codebase and the default hyperparameters. We implement BC based on the TD3+BC repository and re-run it on all datasets. The results of other baselines are taken from the PBRL paper[3] and SPOT paper[2].
>
> **Reference**
>
> [1] Wang et al. Diffusion policies as an expressive policy class for offline reinforcement learning. ICLR 2023.
>
> [2] Wu et al. Supported policy optimization for offline reinforcement learning. NeurIPS 2022.
>
> [3] Bai et al. Pessimistic Bootstrapping for Uncertainty-Driven Offline Reinforcement Learning. ICLR 2022.

---

> > ### Comment · Reviewer_j7HR · 2024-08-11
> >
> > The reviewer appreciates the authors' detailed response and thorough experiments. I have no further questions at this stage and have updated my score accordingly. I am inclined toward a positive outcome for the paper.

---

> > > ### Author Response · Authors · 2024-08-11
> > >
> > > Thank you for your positive feedback! We sincerely appreciate the time and effort you have dedicated to reviewing our manuscript. Your comments and questions are highly valuable to us.

---

### Author Rebuttal · Authors · 2024-08-07

### **Global Response**

We thank all the reviewers for taking the time to read our manuscript carefully and for providing constructive and insightful feedback. We are encouraged by the positive comments of the reviewers, such as:

- Important, interesting, and overlooked research topic (Reviewers VgaK/j7HR/fu1b);
- Robust method requiring only one single hyperparameter configuration (Reviewers VgaK/j7HR/VyLC/fu1b), valuable in offline RL applications (Reviewers VgaK/VyLC);
- Solid theory / establishment of a theoretical link between OOD state correction and OOD action suppression (Reviewers j7HR/VyLC);
- Rigorous evaluation methodology, comprehensive assessment, and good empirical performance (Reviewers VyLC/fu1b/j7HR).

Meanwhile, we have been working hard to address the reviewers' concerns and questions and have provided detailed responses to the individual reviews below. We have also attached a **PDF** to this response containing the additional experiment results. Summary of the PDF:

- Performance of SCAS with slight hyperparameter tuning and comparison with additional baselines in Table 1.
- Experimental results of SCAS+CQL, SCAS+IQL, SCAS+TD3BC, and SCAS with Gaussian policy in Table 2.
- Experimental results of SCAS under different dynamics model errors in Figure 1.
- A replotted version of Figure 2 in the paper in Figure 2.

We hope our response could address the reviewers' concerns. We would be more than happy to resolve any remaining questions in the time we have and are looking forward to engaging in a discussion.

---

### Decision · Program_Chairs · 2024-09-25

**Decision:**

Accept (poster)

**Comment:**

All reviewers, including myself as the Area Chair, commend the originality of this submission, particularly its focus on out-of-distribution (OOD) states rather than the more commonly explored OOD actions. The work is grounded in solid theoretical foundations and presents a novel method that is both simple and elegant. While the empirical results only represent an incremental advance, the submission has significant potential to pave the way for future impactful research in the field of Offline Reinforcement Learning. Therefore, I recommend its acceptance.